# Spatial variability and future evolution of surface solar radiation, aerosols and cloud cover over Northern France and Benelux: a regional climate model approach

Gabriel Chesnoiu[1], Isabelle Chiapello[1], Nicolas Ferlay[1], Pierre Nabat[2], Marc Mallet[2], and Véronique Riffault[3]

[1]Université de Lille, CNRS, UMR 8518 - LOA, Lille F-59000, France
[2]UMR 3589 - CNRM, CNRS, Toulouse F-31057, France
[3]IMT Nord Europe, Institut Mines-Télécom, Université de Lille, Centre for Energy and Environment, 59000 Lille, France
**Correspondence:** Isabelle Chiapello (isabelle.chiapello@univ-lille.fr)

**Abstract.** Improving knowledge of current and future spatiotemporal variability of surface solar radiation is essential in the context of climate change and associated environmental and societal issues. Such an evolution will be influenced by changes in both meteorological parameters and atmospheric composition, notably by anthropogenic emissions. We investigate both all-sky and clear-sky surface solar radiation (SSR) variability, along with cloud cover, aerosols and water vapor content, over Northern

France and Benelux. This region of Europe is largely influenced by cloudy conditions and anthropogenic aerosols, especially nitrate species. Our analysis relies on the CNRM-ALADIN64 regional climate model at 12.5 km x 12.5 km spatial resolution, which includes the TACTIC interactive aerosol scheme. First, hindcast reanalysis-driven simulations over 2010-2020 allow a regional evaluation of ALADIN outputs and to investigate recent spatial variability of SSR and associated atmospheric parameters. Secondly, their possible evolution at mid and end of the 21st century are investigated based on ALADIN climate

simulations following two contrasted CMIP6 scenarios, namely the shared socioeconomic pathways (SSP), SSP1-1.9 and SSP3-7.0. Our regional evaluation of clear-sky and all-sky SSR, clear-sky frequency and aerosols over northern France and Benelux shows reasonable agreement between 2010-2020 ALADIN hindcast simulations and coincident multi-site ground-based measurements, despite some overestimation of nitrate aerosols in spring and an overall underestimation of organic particles by the model. Focusing on spring and summer seasons, hindcast simulations show maximum of solar radiation around

the southern parts and over sea areas of the region. In addition to latitudinal effects, elevated aerosol loads over Benelux, and high cloud cover over the South West of England reduce the SSR. Compared to 2005-2014 atmospheric conditions, ALADIN mid and long-term simulations for SSP1-1.9 predict a significant reduction of aerosol loads, especially over the Benelux, associated with an increase in future clear-sky SSR but geographically limited all-sky SSR evolution. In contrast, SSP3-7.0 simulations projected pronounced and extended decreases of clear-sky and all-sky SSR over most of the Benelux/northern

France region. The reductions are maximum in spring due to combined effects of higher cloud cover and nitrate aerosol increases over the Benelux starting in 2050. In summer, cloud cover upcoming decreases largely attenuate the reduction of SSR due to aerosols in 2050, while in 2100 this attenuation is offset by strong water vapor increases. Thus, this regional climate model approach highlights seasonally and spatially variable impacts of future anthropogenic aerosol emissions on SSR

evolution over the 21st century. Indeed, over this part of western Europe, cloud cover and water vapor modifications will likely largely contribute to modulate forthcoming aerosol influences.

## 1 Introduction

As shown by available long-term observational records in Europe and various regions of the world, surface solar radiation (SSR) has not been stable over the past decades (Wild et al., 2015) and should not be expected to be steady in the upcoming decades. Both changing climate and variations of clouds and aerosols will influence its future evolution (Nabat et al., 2014), with possible regional specificities that need to be investigated (Persad et al., 2023). The quantity of solar energy reaching the Earth's surface plays a critical role in the Earth's energy balance and climate dynamics. It influences a wide range of key physical and biological processes, including evaporation, snow and glacier melt, plant photosynthesis and associated terrestrial carbon uptake (IPCC, 2023). Furthermore, the efficiency of systems relying on the conversion of SSR into other energy forms, such as heat and electricity, directly depends on the availability and variability of the solar resource (Jerez et al., 2015; Tobin et al., 2018; Hou et al., 2021).

In the context of climate change that requires an increase of photovoltaic energy production (relevant for the energy transition), understanding the past and future evolution of the solar radiation incident at the surface is an important environmental and societal issue (Jerez et al., 2015; Tobin et al., 2018). Numerous studies conducted in recent decades, utilizing both observations and modeling, indicate that surface solar radiation has been subject to significant decadal variations in the past, with a worldwide decreasing (dimming) trend until the 1990s, and conversely an increasing (brightening) trend from then onwards (Wild et al., 2005; Wild, 2009; Liepert, 2002; Norris and Wild, 2007). Determining the causes of these trends has been challenging due to the complex interplay of various forcing agents, which directly affect SSR variability through scattering and absorption, and also alter atmospheric dynamics and cloud formation. In Europe, an increasing number of studies suggest that the rise in all-sky radiation since around 1985 is attributable to changes in cloud cover and anthropogenic aerosol emissions (Schwarz et al., 2020; Boers et al., 2017; Dong et al., 2022; Wild et al., 2021). However, it should be noted that due to the high spatial and temporal variability of clouds and aerosols, their influence and the resulting trends in SSR, exhibit strong spatiotemporal variations that require further investigation. Pfeifroth et al. (2018) notably show that, over 1992-2015, maximum positive trends occur in spring and autumn across western, central and eastern Europe, whereas the winter season and southern Europe exhibit weaker increases or even negative trends in SSR. Regarding future evolutions, several studies conducted in Europe, based either on global or regional climate models, project significant changes in meteorological parameters by the end of the 21st century (Coppola et al., 2021), although some uncertainties still persist in the projections associated to CMIP (Coupled Model Intercomparison Project) future climate scenarios. These uncertainties are often related to the type of modeling in terms of regional versus global climate model (Coppola et al., 2021) and robust assessment of aerosols' future evolutions in time (Boé et al., 2020; Gutiérrez et al., 2020; Schumacher et al., 2024). Based on most recent future climate scenarios used in the sixth phase of the Coupled Model Intercomparison Project (CMIP6), Drugé et al. (2021) underline the role of aerosols on the evolution of SSR over the Euro-Mediterranean region, both in terms of aerosol-radiation and aerosol-cloud interactions, while

Hou et al. (2021) emphasize the influence of cloud cover changes over Europe. Both studies highlight some SSR increases over large parts of Europe for several shared socioeconomic pathways (SSP).

The aim of our study is to contribute to refine the current knowledge of the recent and future spatiotemporal variability of surface solar radiation in link with the evolution of associated atmospheric components, focusing on the Benelux and Northern France (BNF) region, a part of Western Europe subject to frequent cloudy conditions and regular atmospheric particulate pollution events (Potier et al., 2019; Favez et al., 2021; Velazquez-Garcia et al., 2023). In order to provide further understanding, our analysis encompasses both all-sky and clear-sky global surface solar radiation, as well as the proportion of diffuse irradiance, which impacts the performances of some solar technologies (Kirn et al., 2015; Lindsay et al., 2020) as well as vegetation productivity and photosynthetic processes (Roderick et al., 2001; Mercado et al., 2009; Niyogi et al., 2004). The variability of all these radiation parameters is examined along with those of three main atmospheric parameters, i.e., cloud cover, atmospheric aerosols and water vapor content, which have a strong influence on global, direct and diffuse SSR levels, and need to be properly accounted for in modeling exercises both for the current period and for future climate trajectories. Our study relies on high-resolution (1-hour model outputs and a spatial resolution of 12.5 x 12.5 km) simulation datasets from the CNRM-ALADIN64 regional climate model, that incorporates the recently upgraded TACTIC interactive aerosol scheme with seven main aerosol species, including ammonium and nitrate particles (Drugé et al., 2019, 2021). The crucial and increasing role of nitrate aerosols in terms of regional climate has been underlined by several studies (Bellouin et al., 2011; Colette et al., 2013; Hauglustaine et al., 2014; Drugé et al., 2019, 2021), especially over western Europe. The regional ALADIN simulations are analyzed both in hindcast (driven at the lateral boundaries by the ECMWF-ERA5 reanalysis, covering years 2010-2020) and climate (driven at the lateral boundaries by the global climate model CNRM-ESM2-1) modes. The regional climate ALADIN simulations allow the study of the future evolution of SSR components (clear-sky, all sky and diffuse fraction) and associated atmospheric parameters (cloud cover, aerosols, water vapour) for the mid-term (2045-2054) and long-term (2091-2100) horizons, compared to the recent historical period (2005-2014). The future simulations are performed for two contrasting CMIP6 scenarios that use the most recent and more realistic anthropogenic emission datasets, which are supposed to improve the representation of the long-term variability of aerosols (Drugé et al., 2021). We choose shared socioeconomic pathways SSP1-1.9 and SSP3-7.0, as they present contrasted climate evolutions, especially for aerosols and precursors emissions, and are thus representative of a large range of possible future scenarios.

Section 2 provides a brief overview of the CNRM-ALADIN64 climate model and the two kinds of simulation datasets, hindcast and climate versions, used in this study. This section also introduces the various sets of ground-based measurements of aerosol loads and SSR components used to evaluate the performances of CNRM-ALADIN64 hindcast simulations over the BNF region for the period 2010-2020, as detailed in Section 3. The results of our analysis of the recent and future spatiotemporal variability of the solar environment over the north of France and Benelux regions are presented in Section 4. In order to allow the evaluation of the influence of aerosol anthropogenic emissions future trajectories on SSR evolution, we focus on spring (i.e. March-April-May) and summer (i.e. June-July-August) seasons. Over the Benelux/Northern France region, both aerosol loads and SSR are maximum at this period of the year, coincidently with more frequent clear-sun conditions, as shown by

ground-based measurements analysis (Chesnoiu et al., 2024b). Finally, Section 5 provides the concluding remarks, including potential directions for future works.

## 2   Models and observations

### 2.1   The CNRM-ALADIN64 regional climate model

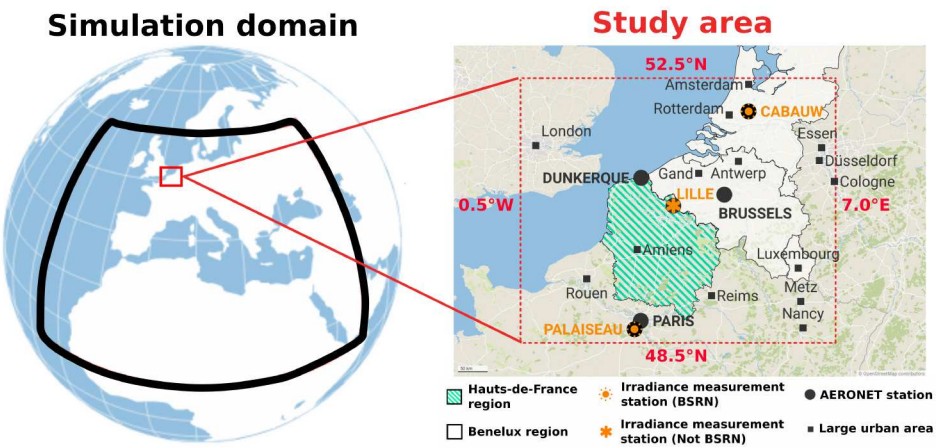

**Figure 1.** Map representing the domain of simulation (black square on the left hand side) as well as the study area (red square) considered in this work. The Hauts-de-France and Benelux region are also represented on the right hand side of the figure, as green hatched and white areas, respectively. The location of the different measurements stations used to evaluate ALADIN HINDCAST simulations are also depicted. AERONET stations equipped with sunphotometers are represented as black dots. An additional orange symbol indicate that coincident irradiance measurements are also available at the station. Large urban areas located in the study area are also represented as small black squares. Maps created with Datawrapper. ©OpenStreetMap contributors 2024. Distributed under the Open Data Commons Open Database License (ODbL) v1.0

In this study, we use regional climate simulations from the most recent version (v6.4) of the CNRM-ALADIN (National Centre for Meteorological Research - Limited Area Adaptation Dynamic International Development) model (Nabat et al., 2020), which includes an interactive aerosol scheme (Drugé et al., 2019, 2021).

CNRM-ALADIN64 is a bi-spectral regional climate model (RCM) with a semi-Lagrangian advection and a semi-implicit scheme, that provides regional simulations over several areas such as the Euro-Mediterranean region considered in the present study (Fig. 1), as shown in the previous studies of Drugé et al. (2019, 2021) and Nabat et al. (2020). This model provides
outputs at monthly, daily and hourly frequencies. The horizontal resolution ranges from 0.44° (50 km) to 0.11° (12.5 km). In the present study, only hourly outputs were analyzed to isolate daytime conditions, which are more relevant for the study of surface solar radiation. Moreover, the horizontal resolution was chosen at 0.11° to optimise comparisons with ground-based measurements as described in Section 3.

The surface is represented by the SURFEX (SURFace EXternalisée, Masson et al. (2013)) continental surface modeling platform version 8, which separates natural land surfaces, lake areas, and maritime zones in surface irradiance calculations. SURFEX provides, in particular, the upward fluxes, latent and sensible heat, and surface albedo required for the radiative scheme of the CNRM-ALADIN64 model. The radiative transfer simulations of CNRM-ALADIN64 are based, for the solar spectrum, on the radiative scheme of Fouquart and Morcrette, which relies on the use of a 6-band wavelength k-distribution (Fouquart and Bonnel, 1980; Morcrette et al., 2008), while thermal radiances simulations rely on the rapid radiative transfer model (RRTM) of Mlawer et al. (1997). In what follows, the model is more simply referred to as "ALADIN".

### 2.1.1 The TACTIC aerosol scheme

The CNRM-ALADIN64 model includes a tropospheric aerosol scheme named TACTIC (Tropospheric Aerosols for Climate in CNRM-CM), which is based on the GEMS/MACC module (Morcrette et al., 2009) of the European Centre for Medium-Range Weather Forecasts (ECMWF) forecasting model. The TACTIC scheme, described in details in Nabat et al. (2020), simulates the life cycle (emission, transport, deposition) of the five aerosol species (in external mixing way) initially included in the model, i.e. black carbon, organic matter, desert dust, sulfate, and sea salt, as well as that of nitrate ($NO_3$) and ammonium ($NH_4$) aerosols, which were recently included (Drugé et al., 2019).

While a driving model (global climate model (GCM) or reanalysis model) provides atmospheric lateral boundary forcing at a 6 hour frequency for temperature, wind, humidity and pressure, as well as sea surface temperature and sea ice cover at monthly frequency (linearly interpolated into daily values), the long-distance transport of particles is not included in the boundary conditions forcing from global model simulations. However, the CNRM-ALADIN64 simulation domain (Fig. 1) is assumed to be sufficiently large to include the main aerosol sources affecting the Euro-Mediterranean region, including the two main desert dust aerosol sources (i.e., the Sahara and the largest part of the Arabian Peninsula).

Primary emissions of sea salt and dust aerosols are computed based on surface winds, and soil characteristics for dust particles, while anthropogenic and biomass burning emissions of aerosols are based on monthly inventories of precursor gases and primary aerosols. Overall, the TACTIC aerosol scheme uses 16 prognostic variables: (i) six size bins for dust and primary sea salt aerosols (three each), (ii) four size bins for organic matter and black carbon aerosols (one hydrophilic and one hydrophobic bin each), (iii) one bin for sulfate particles and another for its precursors, notably sulfur dioxide ($SO_2$), (iv) two different bins for nitrate aerosols (fine and coarse modes), and (v) two bins for ammonium and ammonia tracers. It is noteworthy that one of the ammonium nitrate precursors, nitric acid ($HNO_3$), is implemented in the model as a monthly climatology, constant over the years, based on CAMS reanalysis data (Inness et al., 2019). However, a sensitivity test conducted by Drugé et al. (2019) revealed that the nitrate concentration is relatively unaffected by the use of a constant or time-dependent nitric acid climatology. Additionally, biogenic secondary organic aerosols (SOA) are included in CNRM-ALADIN64 simulations through the climatology of Dentener et al. (2006), but their formation is not explicitly included in the TACTIC aerosol scheme.

The TACTIC aerosol scheme also accounts for the transport of aerosols inside the domain, the influence of humidity on aerosol optical properties, as well as dry and wet deposition processes (in and below clouds). The TACTIC aerosol scheme considers both the aerosol-radiation interactions (forcing and adjustments) for all aerosols, as well as the aerosol-cloud interac-

tion forcing (cloud albedo effect) only for sea salt, sulfate, and organic aerosols. However, the other aerosol-cloud interactions
(second indirect effect on cloud lifetime) are not included in this aerosol scheme.

The various optical properties of each aerosol species, especially those used as input for the radiative transfer scheme (extinction coefficient, asymmetry parameter, and single scattering albedo), are set for each aerosol type as in Nabat et al. (2013) and Drugé et al. (2019). These different aerosol optical properties are pre-calculated using a Mie code based on the assumption of aerosol sphericity (Ackerman and Toon, 1981), and depend on relative humidity, except for mineral dust and
hydrophobic carbon particles. For more information, including cloud and gas optical properties, as well as the dynamics of the CNRM-ALADIN64 regional climate model, refer to the publication by Nabat et al. (2020) and the references therein.

### 2.1.2   Study area

The ALADIN simulation domain used in this study (Fig. 1), close to that used in (Nabat et al., 2015; Drugé et al., 2019; Nabat et al., 2020; Drugé et al., 2021), includes the official domain of the Med-CORDEX initiative (Ruti et al., 2016), as well as the
two main sources of desert aerosols represented by the Sahara and the Arabian Peninsula. Thanks to the finer spatial resolution of 12.5 km of the version 6.4 of ALADIN used here, we focus on analyzing simulations in a much more restricted area (0.5°E → 7°W; 48.5°N → 52.5°N, Fig. 1), hereafter referred to as the Benelux/Northern France (BNF) region, spanning from Paris to Amsterdam and from London to Cologne. This region, which encompasses several European metropolises (London, Paris, Brussels, and Amsterdam), is characterized by both a high population density, with over 70 million inhabitants, as well as
significant levels of anthropogenic activity, particularly from the residential, industrial, transportation, and agricultural sectors, which result in substantial emissions of air pollutants. The BNF region also experiences a significant influence of maritime aerosols originating from the English Channel and the North Sea, along with a relatively high frequency of cloudy conditions throughout the year. Our study area also includes six measurement stations from the AERONET (Aerosol Robotic NETwork, Holben et al. (2001)) network (namely, Brussels in Belgium, Cabauw in the Netherlands, Dunkerque, Lille, Palaiseau, and
Paris in France) and three surface irradiance measurement sites (Cabauw in the Netherlands, Lille, and Palaiseau in France), including two (Cabauw and Palaiseau) from the Baseline Surface Radiation Network (BSRN, Ohmura et al. (1998)). This allows for a relatively broad regional evaluation of ALADIN simulations.

### 2.1.3   Simulation datasets

As previously stated, ALADIN regional climate model simulations are constrained at the boundaries by global GCM or re-
analysis model simulations. In this study, several datasets of ALADIN simulations were used. A first dataset, hereafter referred to as *HINDCAST*, was used to evaluate ALADIN regional climate simulations, as well as to analyze the spatial variability of the solar environment over the BNF region from 2010 to 2020, which represents the longest and most continuous period of coincident photometric and irradiance measurements available for this region. In this case, ALADIN simulations are driven at the lateral boundaries by the reanalysis ECMWF-ERA5 (ECMWF, 2016). A spectral nudging method (as in Nabat et al.,
2020) is also included to better constrain the large scales in the model (above 850 hPa). In addition, anthropogenic and biomass burning emissions are defined using emission inventories from the CEDS v2021-04-21 release datasets (O'Rourke et al., 2021),

and were adjusted following the methodology of Lamboll et al. (2021) to account for the decrease in anthropogenic activities due to lockdown periods of the year 2020.

Using the climate mode of ALADIN, three additional datasets, hereafter referred to as *HIST*, *SSP119* and *SSP370*, were used to assess the future evolution of the solar environment over the BNF region for two Shared Socioeconomic Pathways (SSP, O'Neill et al. (2017)) from the CMIP6 (Coupled Model Intercomparison Project Phase 6) initiative. These simulations are driven at the boundaries by the global climate model CNRM-ESM2-1 (Séférian et al., 2019). The first dataset corresponds to regional climate simulations over the historical period 2005-2014, while the other two correspond to mid-term (2045-2054) and long-term (2091-2100) future regional climate simulations for the optimistic SSP1-1.9 ("Sustainability—Taking the green road", O'Neill et al. (2017)) and more pessimistic SSP3-7.0 ("Regional rivalry – A rocky road", O'Neill et al. (2017)), respectively. To ensure the statistical robustness of our analysis, *HIST*, *SSP119* and *SSP370* datasets are actually composed of three distinct 10-year members for each period, which are aggregated to form, for each period and scenario, five small climate ensembles of 30 years: one for the historical period, two for the mid-term (one for each scenario), and two for the long-term. Anthropogenic and biomass burning emissions are defined over the historical period through the emission inventories of Hoesly et al. (2018) and van Marle et al. (2017), respectively, while future emissions vary according to the pathways defined by Gidden et al. (2019). The two scenarios used in our study were chosen as they correspond to very contrasting anthropogenic emission pathways. On the one hand, the SSP1-1.9 scenario is characterized by a relatively limited increase of greenhouse gas emissions, which leads to lower levels of global warming, and thus to a reduced increase of the surface temperature and water vapor content. Furthermore, as shown in Figure 2, the SSP1-1.9 scenario is also characterized by the greatest decrease in global aerosol emissions (black carbon and organic carbon) and precursor gases (sulfur dioxide and ammonia) in Europe. On the other hand, the SSP3-7.0 scenario is characterized by an important increase in greenhouse gases emissions, and thus in surface temperature and water vapor content. Moreover, the SSP3-7.0 scenario is characterized by the lowest decrease in anthropogenic emissions of aerosol and precursors (Fig. 2), with even an increase in ammonia emissions compared to the reference period of 2005-2014.

It is worth noting that a preliminary analysis of ALADIN simulations has shown overall overestimates of nitrate mass concentrations over the BNF region, in agreement with previous results of Drugé et al. (2019). Although this overestimation was mainly observed in spring, for consistency, a global reduction factor of 25% has been applied to all monthly ammonia ($NH_3$) emissions, as a precursor of nitrate and ammonium aerosols. The choice of the reduction factor has been the subject of an extensive sensitivity analysis. The retained adjustment factor of 25%, specific to our study, represents a compromise between reducing the overestimation in spring and maintaining realistic nitrate concentrations throughout the rest of the year. It can be emphasized that such corrections are consistent with current uncertainties in ammonia emissions, which remain significant, especially at the local scale (Hoesly et al., 2018).

### 2.2 Ground-based observations

In this study, several datasets of ground-based measurements have been used to evaluate the performances of ALADIN simulations over the BNF region (Fig. 1). Aerosol optical properties simulations (mainly aerosol optical depth and Ångström exponent) are compared with AERONET photometric observations available at six measurement stations (Table 1) with more than

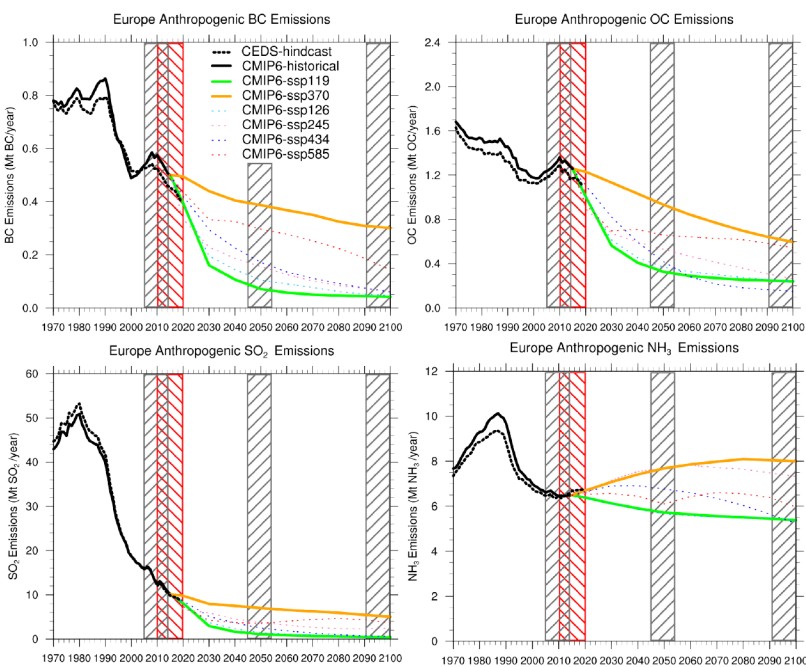

**Figure 2.** Evolution of global aerosol emissions (black carbon and organic carbon, top panels) and precursor gases (sulfur dioxide and ammonia, bottom panels) in Europe from 1970 to 2100 (van Marle et al., 2017; Hoesly et al., 2018; Gidden et al., 2019) for different Shared Socioeconomic Pathways (SSP). CMIP6 historical emissions over the period 1970-2015 are represented by a thick black curves, while projected emissions for the different SSP scenarios appear as colored lines. Emissions for the two scenarios analyzed in this study are represented as thick green (SSP1-1.9) and orange (SSP3-7.0) lines, while other scenarios appear as smaller dashed lines. The grey hatched areas represent the reference period (2005-2014) and future horizons (2045-2054 and 2091-2100) considered for our analysis of the future evolution of the solar environment. Moreover, CEDS emissions (O'Rourke et al., 2021) used for "HINDCAST" simulations are represented by the thick black dashed curves, while the corresponding period (2010-2020) considered for evaluation and analysis of these simulations is highlighted by red hatched areas.

5 years of data over the period 2010-2020 (Brussels, Cabauw, Dunkerque, Lille, Palaiseau, and Paris). Note that AERONET observations also include measurements of integrated PRecipitable Water (PRW), which were used to assess simulations of the water vapor content at all measurement stations. Three of the six stations (Cabauw, Lille, and Palaiseau) also include coincident surface solar radiation (SSR) observations, which were used to evaluate ALADIN radiative transfer simulations of SSR.
In addition, in-situ long-term measurements of $PM_1$ aerosol surface mass concentrations from the ATOLL (ATmospheric Observations in LiLLE) platform in Lille (Chebaicheb et al., 2023) were used to evaluate ALADIN simulations of the aerosol chemical composition near the surface. Note that in what follows only surface mass concentrations, in $\mu g.m^{-3}$, are evaluated, and are therefore mentioned, for simplicity, as surface concentrations.

| Station | Location | Altitude (m) | Number of months (hours) over 2010-2020 | Coincident irradiance measurements |
|---|---|---|---|---|
| Lille (France) | 50.61°N, 3.14°E | 60 | 132 (13 497) | Yes |
| Dunkerque (France) | 50.04°N, 2.37°E | 5 | 115 (9 955) | No |
| Palaiseau (France) | 48.71°N, 2.22°E | 156 | 132 (14 224) | Yes |
| Paris (France) | 48.85°N, 2.36°E | 50 | 124 (12587) | No |
| Cabauw (Netherlands) | 51.97°N, 4.93°E | -1 | 107 (7 488) | Yes |
| Brussels (Belgium) | 50.78°N, 4.35°E | 120 | 117 (9024) | No |

**Table 1.** Station name, location, altitude and number of months and hours available over the study period (2010-2020) of the six AERONET stations considered in this study. The additional availability of irradiance measurements is also indicated.

### 2.2.1 Surface solar radiation

In Lille, a set of a Kipp & Zonen CH1 pyrheliometer and a CMP22 pyranometer (paired with a shading sphere), mounted on a sun-tracking device, measure, respectively, the Direct (or beam) Normal Irradiance (DNI) incident at the surface in the direction of the Sun and the Diffuse Horizontal Irradiance (DHI), respectively. The Beam Horizontal Irradiance (BHI) is then derived from the DNI measurements using the cosine of the solar zenith angle ($\mu_0$) as $BHI = DNI \times \mu_0$. In Cabauw and Palaiseau, which are part of the BSRN network (Driemel et al., 2018; Haeffelin, 2022; Knap, 2022), the total SSR, also known as Global

Horizontal Irradiance (GHI), is measured directly using an additional CMP22 pyranometer (without a shading sphere), while in Lille the GHI is computed as the sum of the BHI and DHI. All measurements are performed at a 1-minute resolution and have been analyzed over the period 2010-2020, which is the most complete period of measurements for all sites considered in this study. According to BSRN requirements, the measurement uncertainties of GHI/BHI/DHI should be of the order of 2/0.5/2% for BSRN stations (McArthur, 2005). However, the study of Vuilleumier et al. (2014) showed that for the direct component,

an uncertainty of 0.5% is probably not achievable in practice, even with the best commercially available technologies. The uncertainties are thus expected to be of around 2 to 3% for the global and diffuse components, and close to 1.5% for the direct component (Vuilleumier et al., 2014). Note that in Lille, the computation of the GHI from measurements of BHI and DHI can lead to measurement biases of the order of ±1.0% (Derimian et al., 2008, 2012). However, as both instruments are subject to regular maintenance, including daily cleanings and periodic calibrations, uncertainties similar to that of Cabauw

and Palaiseau can be expected for irradiance observations performed in Lille. In addition, it is worth noting that in Lille, a measurement gap occurs for winter months, particularly in January and February, as the instruments are regularly sent either in Delft (Netherlands), for a recalibration by the manufacturer, or in M'Bour (Senegal), for calibration of local instruments.

### 2.2.2 Aerosols

The AERONET network uses CIMEL sun/sky photometers to characterize and monitor aerosol optical and columnar micro-
235 physical properties in various locations around the world (Holben et al., 2001). The photometer provides direct measurements

of the Aerosol Optical Depth (AOD) at typically seven wavelengths between 340 and 1020 nm along with their corresponding Ångström Exponents (AE). In addition, PRecipitable Water (PRW in cm) is derived from measurements at 940 nm. In this study, we used level 2.0 (cloud-screened and quality controlled) photometric data from six AERONET measurement stations (Fig. 1, Table 1) available within the BNF region. Note that the evaluation of ALADIN simulations of the AOD was carried at 550 nm. At this wavelength, no measurement is routinely acquired by photometers. Thus, ground-based data of AOD at 550 nm have been interpolated for all AERONET sites considered following the methodology defined by Gueymard and Yang (2020) based on AERONET AOD measurements at 440, 500, 675 and 870 nm. The initial uncertainty of reference AOD measurements is estimated to be of approximately 0.02 for the UV channels (340 and 380 nm) and 0.01 otherwise (Giles et al., 2019). Hence, the uncertainty for $AOD_{550}$ estimates is expected to be of about 0.01 to 0.02. The uncertainty of photometric PRW measurements is expected to be below 15% (Pérez-Ramírez et al., 2014; Smirnov et al., 2004), with an average dry bias of around +5 to +6% (Pérez-Ramírez et al., 2014).

Near-real-time in-situ measurements of surface concentrations and chemical composition of aerosols have been conducted within the ATOLL platform in Lille since 2016 (Chebaicheb et al., 2023). These observations are based on the coincident measurements of a Quadrupole Aerosol Chemical Speciation Monitor (Q-ACSM, Aerodyne Research), and an aethalometer (AE33, Magee Scientific Inc). The ACSM provides data on mass concentrations and mass spectra of five sub-micron non-refractory aerosol species (NR-$PM_1$, Ng et al. (2011)): particulate organics (Org), nitrate, sulfate ($SO_4$), ammonium, and chloride (Cl, negligible here). Additionally, the aethalometer provides, using optical absorption methods equivalent concentrations of black carbon (eBC in $\mu g.m^{-3}$), with an uncertainty estimated in the order of 15%. These measurements, analyzed at an hourly resolution over the period 2016-2020, allow for the assessment of ALADIN simulations of surface concentrations for various types of submicron aerosols, namely, sulfate, nitrate, ammonium, black carbon, and organic particles. For consistency with the terminology of ALADIN simulations, equivalent concentrations of black carbon (i.e. eBC) derived by the aethalometer are hereafter referred to simply as BC. The operational and technical specifications of the instruments are described in more details in Velazquez-Garcia et al. (2023) and Chebaicheb et al. (2023). Reproducibility expanded uncertainties associated to ACSM concentrations have been estimated to be 9, 15, 19, 28, and 36% for NR-PM1, nitrate, organics, sulfate, and ammonium, respectively (Crenn et al., 2015).

## 3 Regional evaluation of ALADIN HINDCAST simulations

The evaluation of ALADIN *HINDCAST* simulations is conducted using ground-based observations available from the stations of BNF region described in Section 2.2. Here, the main objective is to assess the quality of the representation, within ALADIN simulations, of the climatological variability of surface solar radiation and associated main atmospheric parameters, with a focus on sky conditions, aerosols and water vapor content. Section 3.1 focuses on evaluating the monthly variability of the frequency of clear-sky situations, derived from ALADIN cloud fraction (CLT) simulations, through comparison with ground-based estimates based on the surface solar radiation measurements conducted at Cabauw, Lille, and Palaiseau. Subsequently, Section 3.2 presents the results, for all irradiance components (global, direct, and diffuse), of the comparison between

ALADIN SSR simulations with ground-based observations under both all-sky and clear-sky conditions. Section 3.3 outlines
the evaluation of the climatological variability of the aerosol content at all AERONET stations of the BNF region, as well
as a comparison with in-situ measurements of the surface concentrations of submicron particles in Lille. Finally, Section 3.4
investigates the ability of ALADIN to accurately simulate the inter-annual variability of the clear-sky frequency, all-sky surface
solar radiation and aerosol optical depth. It is noteworthy that this analysis relies solely on daytime measurements and hourly
simulations, which are then averaged to the monthly resolution. This approach is necessary, as photometric observations and
identification of clear-sky conditions from SSR measurements can only be achieved during daylight hours, between sunrise
and sunset.

## 3.1 Assessment of the clear-sky frequency

### 3.1.1 Identification of clear-sky situations

Although numerous methods have been described in the literature (Reno and Hansen, 2016; Gueymard et al., 2019; Al Asmar
et al., 2021), our study relies on the well-established method of Long and Ackerman (2000) to distinguish clear and cloudy
situations based on high frequency (3 minutes or less) ground measurements of global and diffuse surface solar radiation.
This method, which has been used for numerous studies (e.g. Elias et al. (2024)), was chosen for its limited number of input
parameters (solar zenith angle, global and diffuse SSR) and high versatility, as it automatically adapts to the specific conditions
of any observational station equipped with measurements of both global and diffuse horizontal irradiances. Our choice is also
based on the results of the comparative study of Gueymard et al. (2019), which showed its high precision for the identification of
clear-sky conditions. The method notably achieved the second lowest "false positive" score (i.e. percentage of cloudy situations
identified as clear-sky) of 7.25%, despite not depending on collocated photometric measurements or clear-sky simulations. The
method of Long and Ackerman (2000) relies on a series of four complementary tests. The aim of the first two tests is to remove
obvious cloudy situations characterized by extreme values of the normalized SSR (test 1) and measured diffuse irradiance
(test 2) through the definition of threshold values. The third and fourth tests are more elaborate as they rely on the analysis of
the temporal variability of the global radiation (test 3) and the normalized diffuse irradiance ratio (test 4), respectively, which
allows the detection of more subtle cloud covers. This method follows an iterative process, which allows an automatic tuning
to the specific conditions of any observational station equipped with measurements of both global and diffuse irradiances. Note
that for this study, the initial values (before iteration) of all parameters involved in the various tests were defined according to
the recommended values reported by Long and Ackerman (2000), and that similar values were obtained for each measurement
station after the iteration process.

The identification of clear-sky ALADIN simulations relies on the definition of a cloud fraction criterion below which a
simulation is categorized as clear-sky. This threshold has been determined for each station through evaluation of the average
cloud fraction simulated by the model during coincident hours identified as clear-sky by the cloud-screening algorithm applied
to the ground observations of Cabauw, Lille and Palaiseau. Similar criteria were derived for all three stations, with mean CLT
values of 3.4, 3.2 and 3.2%, in Cabauw, Lille and Palaiseau, respectively. Therefore, a fixed criterion of 3.5% was chosen in

this study to identify hourly climate simulations which can be considered as clear-sky within each pixel of the study area. It should be noted that the comparison of SSR simulations with and without clouds (cloud-free simulations) for hourly simulations identified as clear-sky indicates a negligible influence of clouds during such occurrences, with discrepancies between simulated surface solar irradiance values with and without clouds of only a few watts per square meter (not shown).

### 3.1.2 Evaluation of the annual cycle

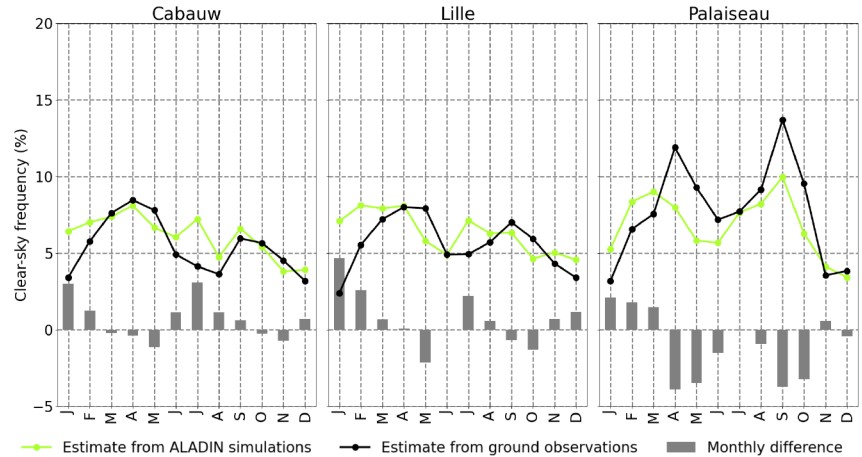

**Figure 3.** Comparisons of the monthly means (2010-2020) of daytime average clear-sky frequency for the three irradiance measurements stations of the BNF region. Green lines correspond to estimates based on ALADIN hindcast simulations (CLT < 3.5%), while black lines represent estimates based on the cloud-screening algorithm of Long and Ackerman (2000) applied to the surface solar irradiance measurements of Cabauw, Lille and Palaiseau. The monthly mean differences between the two kinds of estimates are represented as grey columns.

Figure 3 illustrates the variability of the monthly average daytime frequency of clear-sky situations (in %) over the period 2010-2020 for the three flux measurement sites located in the BNF region (Cabauw, Lille, and Palaiseau). The black curves represent monthly estimates based on the application of the method of Long and Ackerman (2000) to the irradiance measurements at a 1-minute resolution at these different sites. The green curves correspond to estimates based on hourly cloud fraction simulations of the ALADIN regional climate model, for which we consider a simulation as clear-sky if the associated CLT is less than 3.5%. The monthly mean differences between the two kinds of estimates are represented as grey columns.

Estimates based on irradiance measurements show significant intra-annual variability in the estimated clear-sky frequency for the three measurement sites, with a minimum in November-December-January (less than 5%). Maximum clear-sky frequencies occur at two seasons, with a first peak in April (8% in Lille and Cabauw, 12% in Palaiseau) and a second one in September (6-7% in Lille and Cabauw, 14% in Palaiseau). These results highlight the high frequency of cloudy conditions over the whole BNF region. Figure 3 suggests that ALADIN simulations reasonably reproduce the average annual cycle, with a well-represented peak in spring (8%), and a less prominent second peak in September (7%) in Lille and Cabauw. In Palaiseau, the comparison shows that ALADIN slightly underestimates the peaks of clear-sky conditions in September (10%) and in April

(differences in the order of 4% as reported Figure 3), with the first peak occurring earlier (in March, about 9%). In contrast, the model tends to significantly overestimate the frequency of clear-sky conditions in January and February for all sites (absolute differences of around 2 to 5%, or 60-200% in relative terms), although those associated to November and December months are in better agreement (differences less than 1%). Since the average zenith angle is generally higher, one possible explanation for this overestimation could be related to the performance of the initial method of Long and Ackerman (2000), which tends to

misidentify clear-sky situations at large zenith angles (Long and Ackerman, 2000). Additionally, ALADIN estimates show a peak of clear sky frequency in July of about 7% in Lille and Cabauw, which is not recorded by ground-based estimates. Overall, despite some specific differences, these comparisons highlight that ALADIN correctly represents the mean seasonal variations of clear-sky conditions as measured over the BNF region. Moreover, a deeper analysis of our ground-based derived clear-sky conditions shows that the monthly frequency of completely clear-sky hours is significantly lower than the frequency of clear-

sky conditions based on the 1-minute estimates. Such differences, due to the to the highly variable nature of the cloud cover at sub-hourly rates, could explain some of the underestimates of hourly ALADIN simulations, as in September in Palaiseau.

### 3.2   Simulated surface solar radiation

Figures 4a-b depict the average monthly variability over the period 2010-2020 of the measured (colored lines) and simulated (columns) surface solar radiation for the three measurement sites within the BNF region. Panel (a) corresponds to all-sky

conditions, while panel (b) represents irradiances under clear-sky situations identified either from the method of Long and Ackerman (2000) for measurements, or from the cloud fraction filter for simulations.

    The seasonal cycle is characterized by maximum values of all-sky SSR in summer (around 350 $W.m^{-2}$) and a minimum in winter (about 100 $W.m^{-2}$) for all sites, in link with the seasonality of the solar geometry in the region, and additional influence of the clear-sky frequency. Thus, the winter (December-January) minimum of all-sky SSR overlaps with minimum occurrence

of clear-sky conditions, as shown in Figure 3. These influences of the solar geometry and cloud cover also result in high and significantly variable proportions of diffuse irradiance, with average values of diffuse ratio ($R_{diff} = \frac{DHI}{GHI}$) of around 68% in winter and 52% in spring/summer for all sites. This seasonal variability in all-sky SSR and direct/diffuse partition is well simulated by ALADIN, with maximum surface solar radiation values around June-August and a minimum around December, as well as an average contribution of diffuse radiation around 57%, compared to 53% according to measurements. The model

slightly overestimates the diffuse component of SSR (and underestimate the direct one), especially between March and June in Lille and Cabauw, and March and September in Palaiseau, with differences reaching 20 to 30 $W.m^{-2}$ in March-April. These discrepancies may be related to various factors. One possible explanation could be linked to the circumsolar region, an area around the Sun which is measured by instruments as direct radiation but simulated as diffuse flux, and can lead to differences of up to 3% (Gueymard, 2001, 2010; Blanc et al., 2014). The differences may also be related to the simulations of aerosol

and cloud optical depths, as well as cloud fraction. The evaluation of aerosol optical depth simulations is presented in the next section; however, the absence of a similar behavior under clear-sky conditions (Fig. 4b) suggests a minor influence of aerosols, and that differences in all-sky SSR are mainly linked to cloud properties, which unfortunately cannot be evaluated using current available ground-based measurements. It is worth mentioning that the $\delta - Eddington$ approximation is used to

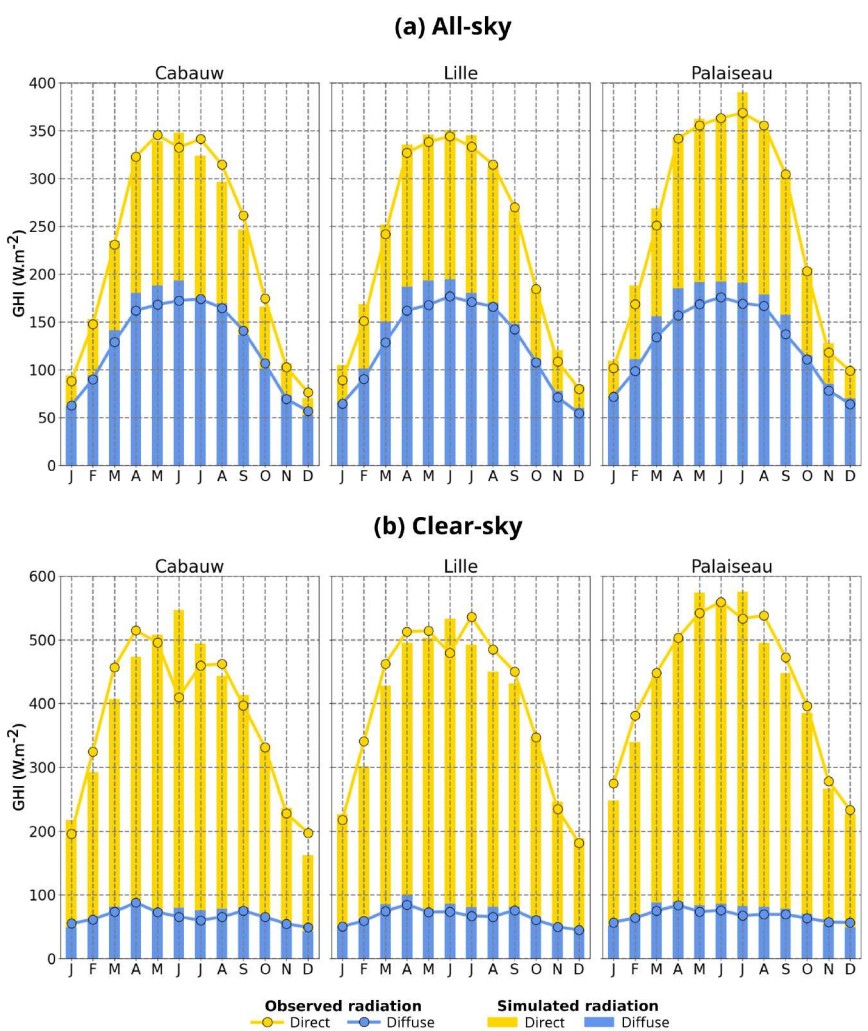

**Figure 4.** Comparisons of the monthly means (2010-2020) of measured (lines) and simulated in hindcast mode (columns) daytime average (a) all-sky and (b) clear-sky irradiances for the three irradiance measurements stations of the BNF region. The total height of each column or line represents the monthly average global irradiance, which is the sum of the direct (in yellow) and diffuse (in blue) components.

simplify the modeling of the cloud phase function in ALADIN simulations, and to reduce the computational burden. Although this approximation does not impact simulation of the global SSR, it leads to an overestimation of the direct component and underestimation of the diffuse irradiance, which may compensate the influence of the other parameters.

Figure 4b shows that the model also seems to accurately reproduce the monthly variability of the global, direct and diffuse SSR under clear-sky conditions, with a maximum around May-June of about 500-550 $W.m^{-2}$ for all sites. However, comparisons in Lille and Cabauw (left and middle panels) show that ALADIN simulations tend to underestimate the direct component in spring by around -20 to -40 $W.m^{-2}$. The parallel analysis of AOD simulations (described in the next section), water vapor content (Fig. S1), and solar zenith angle (Fig. S2) suggests opposite influences of the different parameters. Indeed, in spring, the model tends to overestimate the total aerosol optical depth, resulting in an underestimation of the direct component and an overestimation of diffuse radiation. The parallel overall overestimation of the water vapor content (Fig. S1) amplifies the influence of aerosols on the direct component, while, conversely, mitigating the overestimation of diffuse radiation. Moreover, discrepancies in the occurrence of clear-sky conditions, which vary within the day or month, lead to overall higher solar zenith angle values for clear-sky situations identified from ALADIN simulations between February and April (Fig. S2). This tends to attenuate the overestimation of diffuse radiation due to aerosols, while it further accentuates the underestimation of the direct component. The combined effects of clear-sky occurrences, aerosols and water vapor thus result in a significant underestimation of direct radiation, while conversely, their effects on the diffuse component tend to balance out, leading to a good estimation of diffuse radiation. It should be noted that conversely, in summer, ALADIN clear-sky SSR simulations appear to overestimate all three irradiance components in both Cabauw, Lille and Palaiseau. In this case, there is a strong influence of the occurrences of clear-sky situations, which lead to lower SZA values for clear-sky conditions identified from the simulations (Fig. S2), with a maximum difference of clear-sky SZA of -9° in June in Cabauw, and a corresponding overestimation of solar radiation of more than 100 $W.m^{-2}$ for this month.

## 3.3 Aerosol optical depth and surface concentrations

Figure 5 represents the monthly variability of aerosol optical depth at 550 nm simulated by ALADIN for the AERONET sites present within the BNF region (i.e. Brussels, Cabauw, Dunkerque, Lille, Palaiseau, Paris). The variability of coincident measurements is presented in the form of black boxes whose bounds correspond to the first and third quartiles (i.e. 25 and 75%), respectively. Monthly averages are also represented by black lines inside the boxes. Colored areas correspond to ALADIN AOD simulations for each of the 7 aerosol types considered in the aerosol scheme of the model. The individual contributions of each type are accumulated, hence, the total height of the zones represents the monthly average AOD simulated by ALADIN. As photometric measurements are only reliable in the absence of clouds in the direction of the Sun, they may not be representative of all-sky conditions, thus the evaluation of ALADIN AOD simulations is limited to simulations coincident with measured hourly (diurnal) averages. All measurement stations show a maximum AOD in spring (March-April), and a minimum in winter, with mean values of around 0.2 and 0.1, respectively. The measured summer and autumn mean AOD values are intermediate, mostly around 0.15. This seasonal variability is consistent with those shown by previous analyses in Lille (Chesnoiu et al., 2024b), Cabauw and over Europe (Drugé et al., 2019). Figure 5 indicates that the monthly variability of the AOD is well

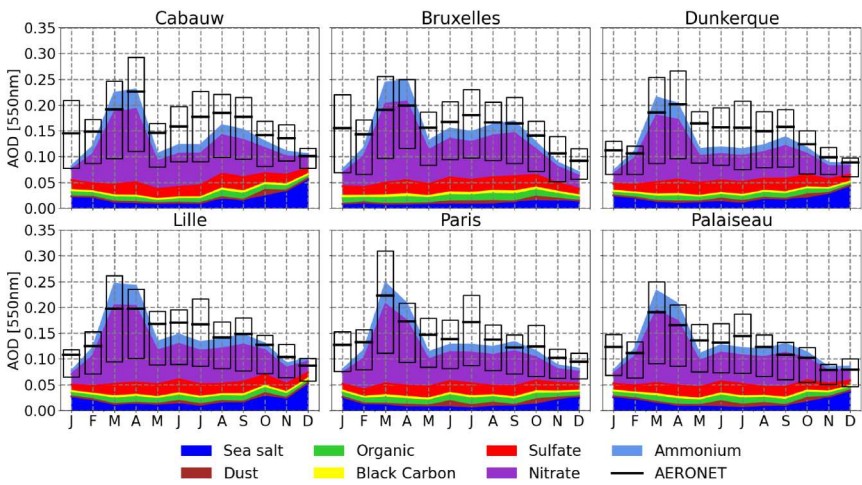

**Figure 5.** Comparisons of the monthly means (2010-2020) of AOD at 550 nm simulated by ALADIN in hindcast mode (colored areas) with coincident photometric measurements (black boxes) for the six AERONET measurement sites located in the BNF region. The boundaries of the boxes correspond to the first and third quartiles (i.e. 25 and 75%), respectively, while the thick black lines inside of each box represent the associated monthly averages.

reproduced in ALADIN simulations for all sites. The simulations suggest that the spring AOD maximum is mostly linked to a significant increase in ammonium nitrate aerosol concentrations, a pattern which is consistent with an increase in agricultural activity at this time of year (Sutton et al., 2013; Roig Rodelas et al., 2019; Van Damme et al., 2022). ALADIN simulations also suggest an increased influence of primary sea salts in winter, particles characterized by low Ångström coefficient values (as reported in Fig. S3). The spring maximum of AOD appears to be often slightly overestimated within ALADIN simulations, by around +0.03 to +0.05, while conversely the model tends to underestimate summer and winter AOD values for all sites. These results confirm those of Drugé et al. (2019), which highlighted a spring overestimation and summer underestimation of the AOD in Cabauw. As discussed in Section 3.2, the overestimation of the AOD in spring could partly explain the underestimation of the direct solar radiation observed under clear-sky conditions (Fig. 4b), while conversely the underestimation of the AOD in summer contributes to the overestimation of the direct component between June and August.

A more detailed analysis of the aerosol chemical composition was carried out by the comparison of the ALADIN simulations with measurements of surface concentrations of $PM_1$ available in Lille over the period 2016-2020 (Velazquez-Garcia et al., 2023; Chebaicheb et al., 2023). Note that ALADIN $PM_1$ concentrations for anthropogenic aerosols (i.e. sulfate, nitrate, ammonium, black carbon and organic matter) are computed from the surface concentrations of the respective bins, following the methodology of Rémy et al. (2019). The results of the comparisons, shown in Figure 6 and Tables 2 (mean seasonal contributions) and S1 (mean seasonal absolute surface concentrations), suggest that the mean seasonal levels of the total surface $PM_1$ concentration (Fig. 6f) are correctly represented by the model in spring and summer, although significantly underestimated in winter and autumn (Tables 2 and S1). Simulations of surface concentrations of sulfate and ammonium aerosols show

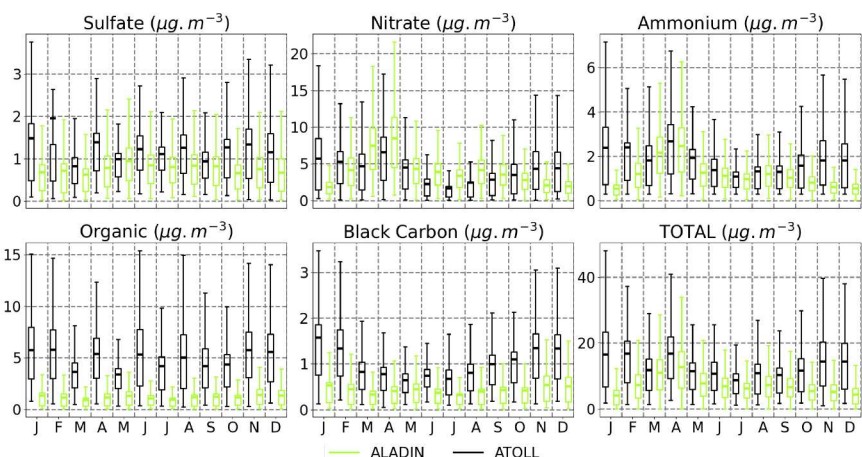

**Figure 6.** Comparison in Lille over the period 2016-2020, under all-sky conditions, of simulated (in green) and measured (in black) monthly mean daytime surface concentrations of $PM_1$ (in $\mu g.m^{-3}$) for 5 types of aerosols: (a) sulfate ($SO_4$), (b) nitrate ($NO_3$), (c) ammonium ($NH_4$), (d) organic aerosols and (e) black carbon. Panel (e) represents the total monthly $PM_1$ concentrations. The colored line inside each box represents the monthly mean value, while the lower and upper ends of the box correspond, respectively, to the first and third quartiles (of height IQR), and the whiskers represent the maximum and minimum values obtained in the interval [median - 1.5IQR; median + 1.5IQR].

| | Sulfate (%) | | Nitrate (%) | | Ammonium (%) | | Organic (%) | | Black carbon (%) | | Total $PM_1$ concentration $(\mu g.m^{-3})$ | |
|---|---|---|---|---|---|---|---|---|---|---|---|---|
| | Model | ATOLL | Model | ATOLL | Model | ATOLL | Model | ATOLL | Model | ATOLL | Model | ATOLL |
| **Winter** | 11.7 | 9.6 | 45.1 | 32.0 | 13.0 | 13.7 | 21.9 | 35.8 | 8.4 | 8.9 | **5.8** | **15.9** |
| **Spring** | 7.1 | 8.0 | 61.4 | 39.4 | 17.8 | 16.0 | 10.1 | 31.0 | 3.6 | 5.6 | **11.0** | **13.3** |
| **Summer** | 11.6 | 11.8 | 53.3 | 20.8 | 15.4 | 12.4 | 14.7 | 47.8 | 5.0 | 7.3 | **7.1** | **10.1** |
| **Autumn** | 12.3 | 9.7 | 46.3 | 29.2 | 13.3 | 12.8 | 20.5 | 39.0 | 7.6 | 9.4 | **6.1** | **12.2** |
| **Overall** | **10.1** | **9.6** | **53.3** | **31.0** | **15.4** | **13.8** | **15.6** | **37.7** | **5.7** | **7.8** | 7.5 | 12.9 |

**Table 2.** Mean contribution (in %), per season and on average over the period 2016-2020, of the different aerosol types to the daytime total surface concentration of $PM_1$ simulated by ALADIN or measured by the ACSM and aethalometer from the ATOLL platform in Lille.

that ALADIN represents quite well the variability of these species both in terms of amplitude and monthly cycle (Figs. 6a and c), with average contributions of the order of 10 and 15%, respectively (Table 2). The overall underestimation of simulated surface mass concentration of $PM_1$ seems to be mostly linked to a systematic underestimation of organic aerosol concentrations throughout the year of the order of -3 to -4 $\mu g.m^{-3}$ (Fig. 6d and Table S1), i.e., about a factor 2. Thus, the contribution
of organic aerosols to surface concentrations represents only 16% in ALADIN simulations, compared to 38% in the ATOLL measurements (Table 2). Moreover, this evaluation shows that surface mass concentrations of black carbon aerosols are also underestimated by ALADIN throughout the year, with monthly differences of the order of -0.5 to -1 $\mu g.m^{-3}$ (Fig. 6e and Table S1), which could lead to an underestimation of the absorbing properties of aerosols in radiative transfer simulations. Nonetheless, as the total concentration of $PM_1$ is underestimated, the overall contribution of black carbon aerosols simulated
by ALADIN remains relatively similar to the measurements, with a contribution of the order of 6 to 8% on average (Table 2). The comparisons shown in Figure 6b highlight that the underestimation of organic and black carbon aerosols is partially offset in spring and summer by a coincident overestimation of nitrate aerosol concentrations, despite the application of a 25% correction factor on ammonia emissions, the main precursor of nitrate aerosols. This offset is especially significant in March and April with differences in total concentrations of around +2 $\mu g.m^{-3}$ between the model and the measurements. As shown
Table 2, the mean annual contribution of nitrate aerosols to the total $PM_1$ concentration is 53% in ALADIN simulations, compared to a mean value of 31% derived from the ground-based measurements. This suggests that the overestimation of the AOD observed in spring is mostly linked to an overly large contribution from nitrate aerosols simulated by ALADIN at this season, while on the contrary the underestimation of the AOD in summer and winter is likely linked to the underestimation of organic and black carbon aerosols. In their study, Drugé et al. (2019) suggest that the underestimation of the AOD in summer
could be linked to the absence of anthropogenic secondary organic aerosols within ALADIN simulations. Note that the recent study of Chebaicheb et al. (2024), based on chemistry-transport simulations from CHIMERE (Menut et al., 2021), similarly highlights a systematic underestimation of organic aerosols across 13 French stations, including ATOLL. These results suggest insufficient understanding of organic aerosol sources or formation process, especially at local scales.

## 3.4 Monthly time series over 2010-2020

Figures 7 compare the simulated and measured monthly times series of daytime (a) clear-sky frequency, (b) all-sky surface solar radiation and (c) aerosol optical depth at 550 nm in Lille over the period 2010-2020. As this period is too brief to assess ALADIN's ability to represent observed trends, this section focuses on the analysis of monthly and seasonal anomalies observed during 2010-2020. Comparisons of aerosol surface concentrations are also excluded as $PM_1$ ground measurements in Lille are only available from 2016 to 2020. For brevity, only comparisons in Lille are presented in this section. Corresponding
figures for Cabauw and Palaiseau are reported in the supplements (Figs. S4 and S5, respectively).

Overall, despite specific seasonal biases highlighted in Sections 3.1, 3.2 and 3.3—such as the overestimation of clear-sky frequency in winter, the underestimation (overestimation) of direct (diffuse) all-sky SSR in March-April, and the overestimation (underestimation) of the AOD in spring (summer)—Figures 7a-c show that ALADIN simulations correctly capture the inter-annual variability of the clear-sky frequency, SSR and AOD. The timing of anomalies is well represented, although the

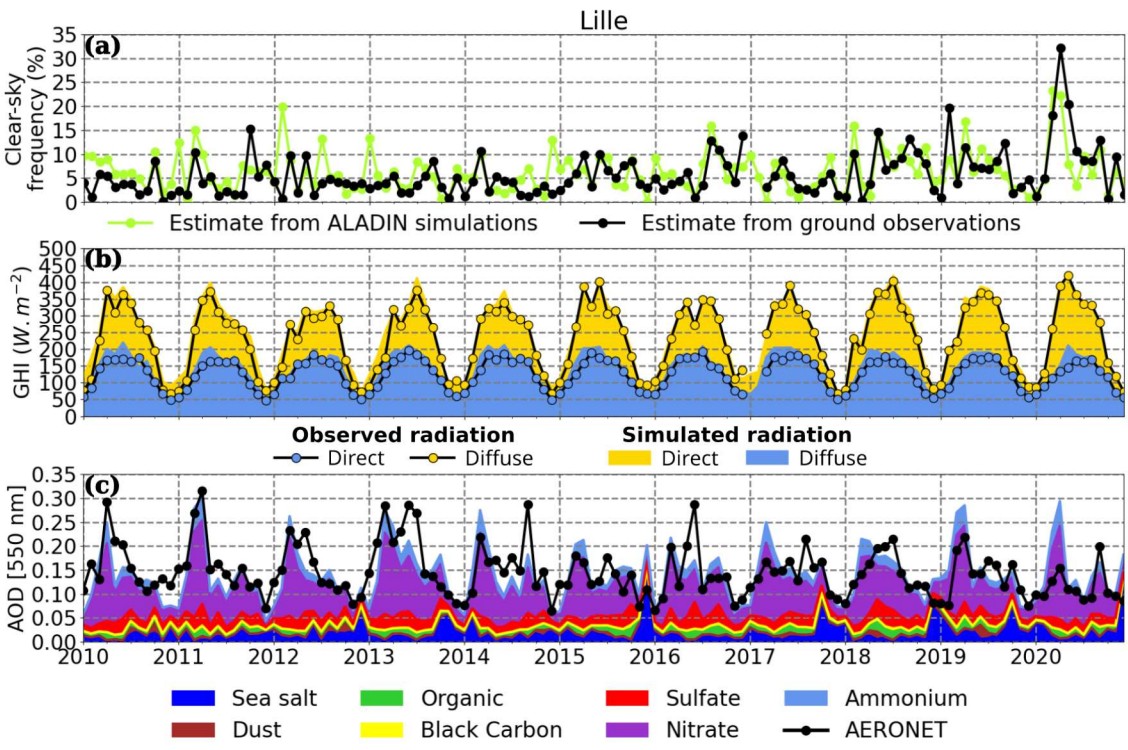

**Figure 7.** Comparisons of the simulated and measured monthly time series of daytime averaged (a) clear-sky frequencies, (b) all-sky irradiances and (c) AOD at 550 nm in Lille over the period 2010-2020. In panel (a), green lines correspond to estimates based on ALADIN hindcast simulations (CLT < 3.5%), while black lines represent estimates based on the cloud-screening algorithm of Long and Ackerman (2000) applied to the surface solar irradiance measurements. In panel (b), ALADIN hindcast simulations and ground-based measurements are represented as colored areas and lines, respectively. The total height of each area or line represents the monthly averaged global irradiance, which is the sum of the direct (in yellow) and diffuse (in blue) components. In panel (c), ALADIN hindcast simulations of the seven aerosol types are represented as colored areas, while AERONET photometric measurements are depicted by thick black lines. As photometric measurements are only reliable in the absence of clouds in the direction of the Sun, they may not be representative of all-sky conditions. Hence, for consistency, monthly mean values derived from ALADIN hourly AOD simulations are limited to simulations coincident with measured hourly (diurnal) averages.

intensity of the extreme values may be slightly biased. For example, the model captures the anomaly in extremely high clear-sky conditions frequency recorded in spring 2020 (more than 20% in Lille, Fig. 7a) and correspondingly high levels of surface radiation (around 400 $W.m^{-2}$, Fig. 7b), which have been previously documented in Lille (Chesnoiu et al., 2024b) and Cabauw (Heerwaarden et al., 2021). Anomalies of AOD at 550 nm are also well simulated by ALADIN, especially the maximum of spring 2011, which is observed for the three measurement stations (Figs. 7c, S4c and S5c). These monthly time series also

suggest that ALADIN overestimations of the AOD in spring seem more frequent over recent years, especially in Lille (Fig. 7c) and Palaiseau (Fig. S5c).

## 4 Present and future surface solar radiation variability

### 4.1 Spatial variability over the period 2010-2020

In this section, we analyze the spatial variability of the SSR and associated atmospheric parameters within the BNF region

over the period 2010-2020, based on ALADIN hindcast simulations for spring and summer. Figures 8a-e illustrate the spatial variability under all-sky conditions of the mean daytime (a) surface solar radiation (SSR), (b) ratio of diffuse irradiance ($R_{diff}$), (c) cloud fraction (CLT), (d) aerosol optical depth (AOD), and (e) precipitable water (PRW), for spring (top row) and summer (bottom row).

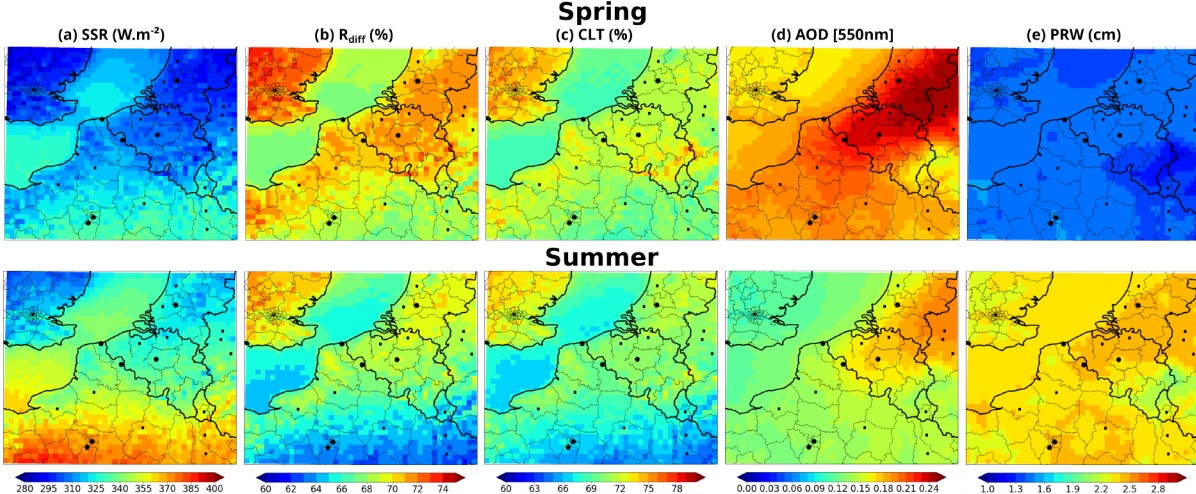

**Figure 8.** Seasonal averages over the period 2010-2020 in spring (upper figures) and summer (lower figures) of ALADIN daytime hourly hindcast simulations under all-sky conditions of (a) surface solar radiation, (b) diffuse ratio ($R_{diff} = \frac{DHI}{GHI}$), (c) cloud fration, (d) AOD at 550 nm, and (e) precipitable water.

ALADIN simulations highlight a strong spatial variability of all-sky SSR over the BNF region, from around 300 $W.m^{-2}$ up

to 400 $W.m^{-2}$ in summer (350 $W.m^{-2}$ in spring). For both seasons, maximum values are simulated over the southern part of the region, due primarily to latitudinal effects on solar geometry, but also over sea surfaces that include the English channel

and North Sea. Minimum values of SSR are simulated over the Benelux and south-west of England. Note that clear-sky SSR simulated by ALADIN reported in Figure S6b show quite similar spatial patterns over the BNF region, with overall higher values, i.e., 450-580 $W.m^{-2}$ in summer, and 400-520 $W.m^{-2}$ in spring. Diffuse fractions simulated by ALADIN over the BNF region vary in the range 60-75% in summer (65%-75% in spring), with maximum values generally associated to minima of SSR, as shown over the northern part and continental areas of the region, especially south-west of England and Benelux (Figs. 8a-b). Interestingly, ALADIN simulations suggest that despite relatively similar incident surface solar radiation levels (around 300-330 $W.m^{-2}$ depending on the season) and proportions of diffuse irradiance (68-74%) over the south-west of England and Benelux regions, these areas are subject to somewhat different atmospheric influences. Indeed, further analysis of cloud fraction simulations (Fig. 8c) suggests that the south-west of England experiences higher levels of cloud fraction, with CLT values of around 73-75% against 69-72% for the Benelux region, which would imply lower SSR (higher $R_{diff}$) values. Conversely, Figures 8d show that the Benelux region displays maximum AOD values, reaching 0.25 in spring (0.20 in summer), while the south-west of England exhibits minimum of AOD within the BNF region, with values consistently lower than 0.15 for both spring and summer seasons. Thus, relatively low values of all-sky SSR over the Benelux may be more linked to high aerosol loads, compared to those simulated over south west England, which seem reduced by the higher cloud cover. Note that the influence of the water vapor content on these radiation patterns is limited as PRW values are relatively homogeneous within the study area, with values ranging from around 1.5 cm in spring to 2.4-2.5 cm in summer (Fig. 8e). However, Figure 8e suggest that ALADIN simulations of the water vapor content are fairly impacted by the orography and vegetation, as PRW values are lower (1.3 cm in spring and 2.1 cm in summer) over the Ardennes region (crossroad between Belgium, Germany and Luxembourg). Interestingly, even at high latitudes, the English channel and North Sea regions exhibit higher levels of SSR and lower proportions of diffuse irradiance than over parallel continental areas, with values close to those simulated over the southern parts of the BNF region (Figs. 8a and b). Over these oceanic areas, SSR ($R_{diff}$) values range from around 320 $W.m^{-2}$ (67%) in spring to about 350 $W.m^{-2}$ (64%) in summer. Figures 8c and S6a suggest that this pattern is mostly linked to the spatial variability of the cloud cover, which is minimal over both the English Channel/North Sea regions and Southern part of the study area, with CLT values of about 68% in spring and 65% in summer. These oceanic areas also exhibit higher frequencies of clear-sky conditions, mostly in the range of 8 to 10% in both spring and summer, in contrast to average values lower than 8% over most of the Benelux and south-west of England regions.

Corresponding simulations of the spatial variability of SSR and associated atmospheric parameters for winter (i.e. December-January-February) and autumn (i.e. September-October-November) seasons are reported in the supplements (Fig. S7). For these two seasons, the spatial patterns are similar to those observed in spring and summer (Fig. 8). However, as expected, AOD ranges are significantly reduced over most of the BNF region, together with increased simulated CLT levels, from 72% to more than 80%. Simulated SSR are largely reduced, below 150 $W.m^{-2}$ in winter, and 220 $W.m^{-2}$ in autumn. Thus, in order to evaluate the impacts of contrasted anthropogenic aerosol future emissions on the high-end range of SSR, we will focus on spring and summer seasons.

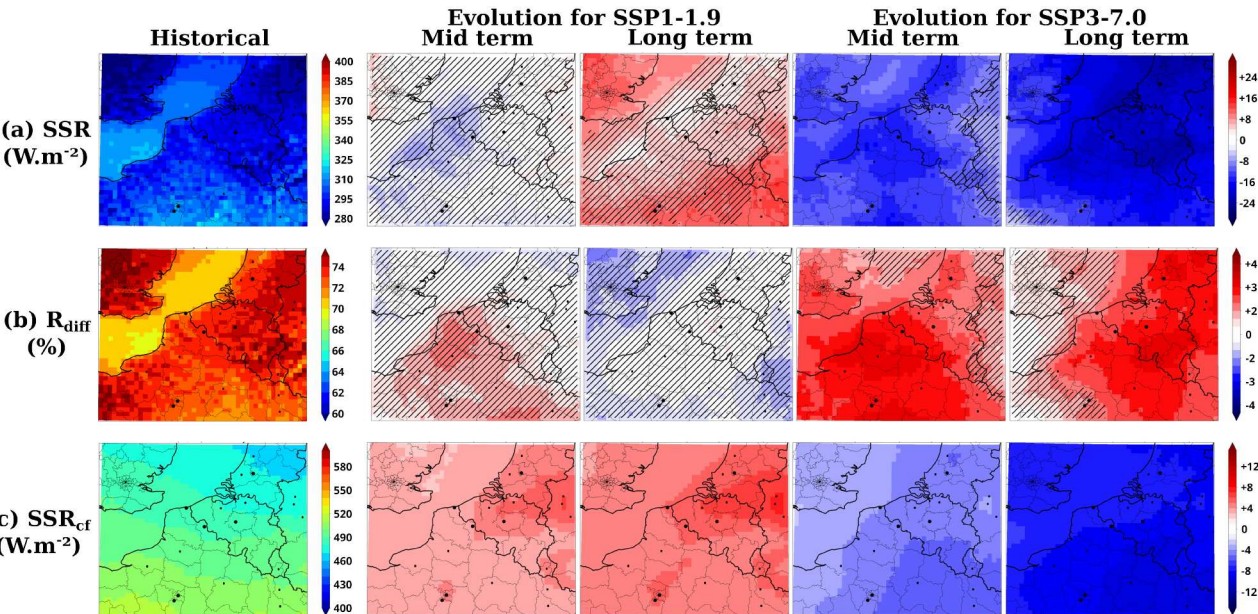

**Figure 9.** Future evolutions in spring for CMIP6 scenarios SSP1-1.9 and SSP3-7.0 of the daytime (a) all-sky SSR, (b) all-sky diffuse ratio ($R_{diff}$) and (c) cloud-free SSR ($SSR_{cf}$) simulated by ALADIN compared to the reference climate simulations over the period 2005-2014 (left panels). The designation "Mid term" represents future changes projected under each scenario for the period 2045-2054, while "Long term" refers to the period 2091-2100. Hatched areas correspond to areas characterized by non-significant changes relative to a Student t-test with a significance level of 10%.

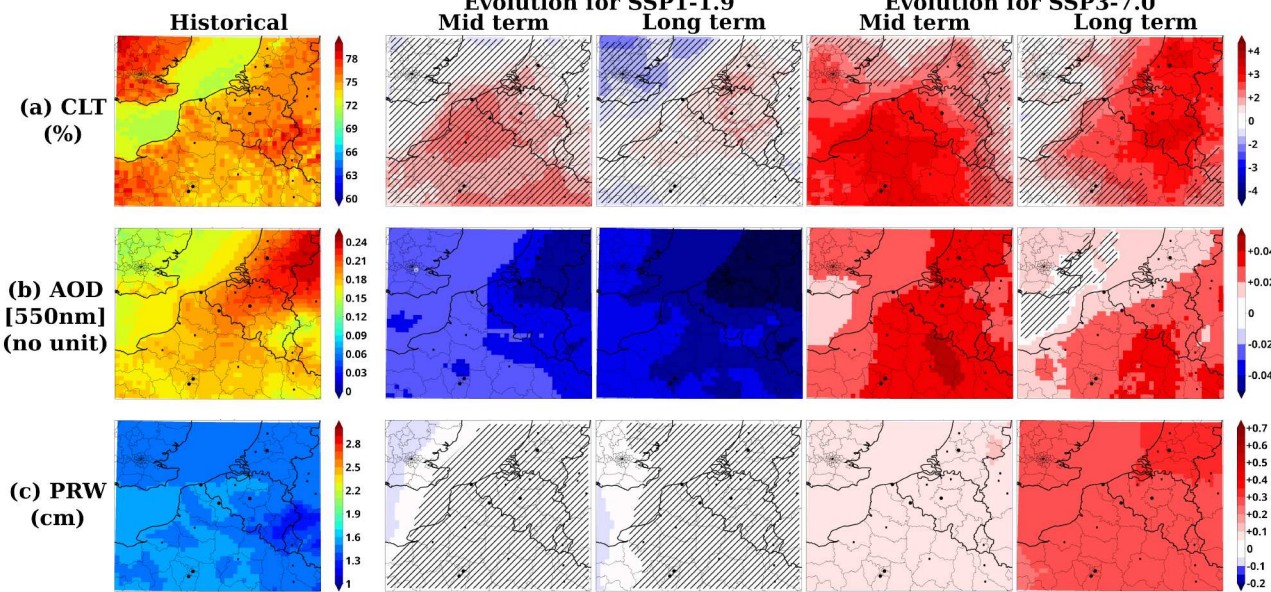

**Figure 10.** Future evolutions in spring for CMIP6 scenarios SSP1-1.9 and SSP3-7.0 of the daytime all-sky (a) cloud fraction, (b) AOD at 550 nm and (c) precipitable water vapor content simulated by ALADIN compared to the reference climate simulations over the period 2005-2014 (left panels). The designation "Mid term" represents future changes projected under each scenario for the period 2045-2054, while "Long term" refers to the period 2091-2100. Hatched areas correspond to areas characterized by non-significant changes relative to a Student t-test with a significance level of 10%.

| Mean aerosol optical depth at 550 nm | | | | | | | | | |
|---|---|---|---|---|---|---|---|---|---|
| | | Total | Sulfate | Nitrate | Ammonium | Organic matter | Black carbon | Sea salt | Desert dust |
| **HINDCAST** | **2010-2020** | 0.19 | 0.026 | 0.097 | 0.026 | 0.009 | 0.003 | 0.027 | 0.004 |
| **Historical** | **2005-2014** | 0.17 | 0.028 | 0.079 | 0.021 | 0.007 | 0.003 | 0.028 | 0.004 |
| **SSP1-1.9** | **2045-2054** | 0.14 | 0.009 | 0.079 | 0.021 | 0.003 | 0.000 | 0.025 | 0.002 |
| | **2091-2100** | 0.13 | 0.006 | 0.074 | 0.016 | 0.006 | 0.000 | 0.025 | 0.002 |
| **SSP3-7.0** | **2045-2054** | 0.20 | 0.017 | 0.110 | 0.030 | 0.006 | 0.002 | 0.031 | 0.004 |
| | **2091-2100** | 0.19 | 0.013 | 0.106 | 0.029 | 0.004 | 0.001 | 0.032 | 0.003 |

(a) Spring

| Mean aerosol optical depth at 550 nm | | | | | | | | | |
|---|---|---|---|---|---|---|---|---|---|
| | | Total | Sulfate | Nitrate | Ammonium | Organic matter | Black carbon | Sea salt | Desert dust |
| **HINDCAST** | **2010-2020** | 0.14 | 0.026 | 0.062 | 0.017 | 0.009 | 0.003 | 0.022 | 0.003 |
| **Historical** | **2005-2014** | 0.13 | 0.029 | 0.054 | 0.014 | 0.008 | 0.003 | 0.019 | 0.002 |
| **SSP1-1.9** | **2045-2054** | 0.11 | 0.009 | 0.060 | 0.016 | 0.005 | 0.001 | 0.018 | 0.002 |
| | **2091-2100** | 0.10 | 0.006 | 0.056 | 0.015 | 0.004 | 0.000 | 0.017 | 0.002 |
| **SSP3-7.0** | **2045-2054** | 0.15 | 0.016 | 0.081 | 0.022 | 0.008 | 0.002 | 0.019 | 0.002 |
| | **2091-2100** | 0.15 | 0.013 | 0.084 | 0.023 | 0.007 | 0.002 | 0.020 | 0.002 |

(b) Summer

**Table 3.** Mean overall AOD at 550 nm over the BNF region and corresponding AOD of the different aerosol types simulated by ALADIN for *HINDCAST*, *HIST*, *SSP119* and *SSP370* datasets in (a) spring and (b) summer.

## 4.2 Future evolution according to two CMIP6 scenarios

Figures 9 represent the future evolution in spring for the two CMIP6 scenarios considered in this study (SSP1-1.9 and SSP3-7.0) of the daytime (a) all-sky SSR, (b) diffuse ratio ($R_{diff}$) and (c) cloud-free SSR ($SSR_{cf}$; i.e. simulations without clouds at all timestamps and not only for clear-sky situations) simulated by ALADIN compared to the historical climate simulations over the period 2005-2014 (left panels). Hatched areas indicate regions characterized by statistically non-significant changes relative to a Student t-test with a significance level of 10%. In addition, Figures 10 illustrate the parallel evolution of the atmospheric content simulated by ALADIN, focusing on the (a) cloud fraction, (b) aerosol optical depth and (c) precipitable water. Note that Figures S10 also depict the long-term (2091-2100) future evolutions of $SSR_{cf}$ and PRW for SSP3-7.0 during both spring and summer, utilizing specific colormaps to enhance the visualization of the projected changes.

Considering the spring season, ALADIN simulations predict a rather weak future evolution of the all-sky SSR for the SSP1-1.9 scenario, which is only significant by the end of the century. For this scenario, ALADIN simulations project spatially limited increases of around +5 to +10 $W.m^{-2}$ over the south-west of England region and south-east parts of the BNF region (Fig. 9a). Conversely, for the SSP3-7.0 scenario, ALADIN predicts more pronounced and widespread decreases of all-sky SSR for this season (Fig. 9a), which are already statistically significant from the middle of the century over most of the BNF region (-10 to -15 $W.m^{-2}$). These large decreases of SSR reach maximum values at the end of the century, with a maximum diminution of around -25 $W.m^{-2}$ over the Benelux region, i.e., 8% decrease compared to the historical period 2005-2014 (leftmost panel of Fig. 9a). Further analysis of ALADIN simulations suggest that the projected springtime decrease in SSR for SSP3-7.0 may be linked to a significant increase in cloud fraction of about +2 to +4% over a large portion of the BNF region, with maximum values around the north of France and Benelux regions (Fig. 10a). Note that the increase in cloud fraction coincides with a decrease in the frequency of clear-sky situations of around -2 to -3% over the Hauts-de-France and Benelux regions (Fig. S8a).

ALADIN simulations for SSP3-7.0 predict a significant increase of the AOD over the BNF region in spring, with values ranging from about +0.01 to +0.05, and maximum changes over the north of France and Benelux regions, both for 2050 and 2100 (Fig. 10b). This increase is mostly related to a rise in ammonium nitrate aerosols, which compensates a decrease in sulfate, organic and black carbon aerosols (Table 3a). As shown Table S2a, these evolutions cause a near to 10% increase of the contribution of nitrate aerosols to the total AOD under SSP3-7.0, rising from 46.5% to 55-56%. These results are consistent with the projected changes in aerosol and their precursor emissions in Europe for the SSP3-7.0 scenario, with a significant increase in ammonia emissions up to 2100 and a slow decrease in other species, notably $SO_2$, BC and OC (Fig. 2). The results of the sensitivity study of Chesnoiu et al. (2024b) reported in Table S3 allow the quantification of the average impact of an increase in AOD on cloud-free SSR over the BNF region when multiplied by the future changes in AOD projected by ALADIN. This approach suggests that the increase of around +0.01 and +0.05 in AOD simulated by ALADIN in spring under SSP3-7.0 would induce a limited decrease in SSR (in the absence of clouds) of around -2 to -8 $W.m^{-2}$.

ALADIN spring simulations for SSP3-7.0 also highlight a rather homogeneous increase of PRW of around +0.1 in 2050 and +0.3 cm in 2100 (Fig. 10c). These increases in PRW are related to coincident rises in simulated surface temperatures, especially at the end of the century (+1.5 to +2°C, Fig. S9a). Following the approach described previously for the AOD based on the results of the sensitivity study of Chesnoiu et al. (2024b) reported in Table S3, we find that the range of future PRW evolution simulated by ALADIN would lead to an additional decrease in cloud-free SSR of about -2 to -5 $W.m^{-2}$.

Consistently, the overall decrease of $SSR_{cf}$ simulated by ALADIN for SSP3-7.0 in spring reach -3 to -7 $W.m^{-2}$ in 2050 and -8 to -12 $W.m^{-2}$ in 2100 (Fig 9c), reflecting the combined effects of aerosols and water vapor. Interestingly, our springtime simulations for SSP3-7.0 show stronger decreases of all-sky SSR (compared to cloud-free SSR), due to the additional contribution of increasing cloud fraction over most of the BNF region (right panels of Figure 9a and 10a). In addition, the increase in CLT, and to a lesser extent in AOD, also results in a fairly significant increase in the proportion of diffuse irradiance , which can reach +2 to +4% over large areas covering the north of France and Benelux regions (Fig. 9a).

In contrast, ALADIN simulates, under SSP1-1.9, a pronounced and widespread decrease in springtime AOD in the range -0.02 to -0.05, with a maximum in 2100 over the Benelux area (Fig. 10b). These decreases are consistent with the projected changes in aerosols and their precursor emissions over Europe under SSP1-1.9 (Fig. 2). Figures 10c also show that the future evolution of PRW in spring is not significant for this scenario due to weaker changes in surface temperature for this scenario (Fig. S9a). Therefore under SSP1-1.9, ALADIN simulates moderate increases in cloud-free SSR in spring, driven mainly by the decrease in AOD, with values ranging from +2 $W.m^{-2}$ to +8 $W.m^{-2}$, reaching a maximum by the end of the century over the Benelux region (Fig. 9c). The future evolution of the cloud fraction simulated by ALADIN for SSP1-1.9 in spring is generally not significant (Fig. 10a). Hence, the future evolution of all-sky SSR is mostly not significant under SSP1-1.9 in spring, except for some specific areas, such as the south-west of England characterized by a decrease in CLT (even if not significant) and in AOD, which show a moderate increase in all-sky SSR by the end of the century (Fig. 9a).

Regarding the projections of ALADIN in summer, Figures 11a-c and 12a-c represent the future evolution of the surface solar radiation and associated key atmospheric parameters during daytime (all-sky and cloud-free SSR, $R_{diff}$, CLT, AOD, PRW) for the two CMIP6 scenarios considered in this study (SSP1-1.9 and SSP3-7.0). Overall, as in spring, the model projects an increase in AOD for SSP3-7.0 of around +0.01 to +0.04, with a maximum over the south-east of the region (Fig. 12b), mainly due to an increase in ammonium nitrate aerosols (Table 3b). Moreover, due to more pronounced changes in surface temperature (+2 to +3°C, Fig. S9b), ALADIN simulates a greater increase in water vapor than in spring, reaching values of up to +0.7 cm by the end of the century over the whole BNF region (Fig. 12c). Consequently, for SSP3-7.0, the combined effects of increasing aerosols (-2 to -7 $W.m^{-2}$ derived from Table S3) and water vapor (-2 to -12 $W.m^{-2}$ from Table S3) result in an overall pronounced summertime decrease in cloud-free surface solar irradiance of around -3 to -13 $W.m^{-2}$, with a maximum over the south-east of the BNF region (Fig. 11c). Conversely, for the SSP1-1.9 scenario, ALADIN predicts a decrease in summertime AOD comparable, although slightly less intense, to that described in spring (Fig. 10b), with values ranging from -0.01 to -0.04 and a maximum centered over the eastern part of the BNF region (Fig. 12b). However, in contrast to the spring season, ALADIN simulations also project an increase of around +0.1 to +0.2 cm in summertime water vapor content (Fig. 12c). As shown Table S3, such an increase in PRW is expected to at least partly attenuate the effect of decreasing aerosols (+2 to +7 $W.m^{-2}$ from Table S3) by around 2 to 3 $W.m^{-2}$, resulting in rather small changes in $SSR_{cf}$, which are only significant in 2100 (+2 to +5 $W.m^{-2}$, Fig. 11c).

According to scenario SSP1-1.9 in 2100, ALADIN simulates greater and more statistically significant decrease in cloud fraction in summer (Fig. 12a) than in spring (Fig. 10a), especially over the south-west of England and North Sea. Hence, this decrease in cloud cover combined with the reduced impact of decreasing aerosols, leads to a future evolution of all-sky SSR fairly comparable to that described in spring. Indeed, Figures 11a show an increase of around +10 $W.m^{-2}$ for 2100, spatially limited over two areas, south west England and the eastern part of the BNF region, as well as a significant decrease in the proportion of diffuse SSR over the northern parts of the BNF region (-1 to -2%, Fig. 11b). Interestingly, according to the SSP3-7.0 scenario, ALADIN also projects a decrease in cloud fraction in summer of around -2 to -3% (opposite to the spring increase shown Figure 9a), which is mostly significant over the northern and/or south-west parts of the BNF region (Fig. 12a). Such a decrease tends to mitigate the combined effects of increased aerosols and water vapor on all-sky SSR changes, leading to a

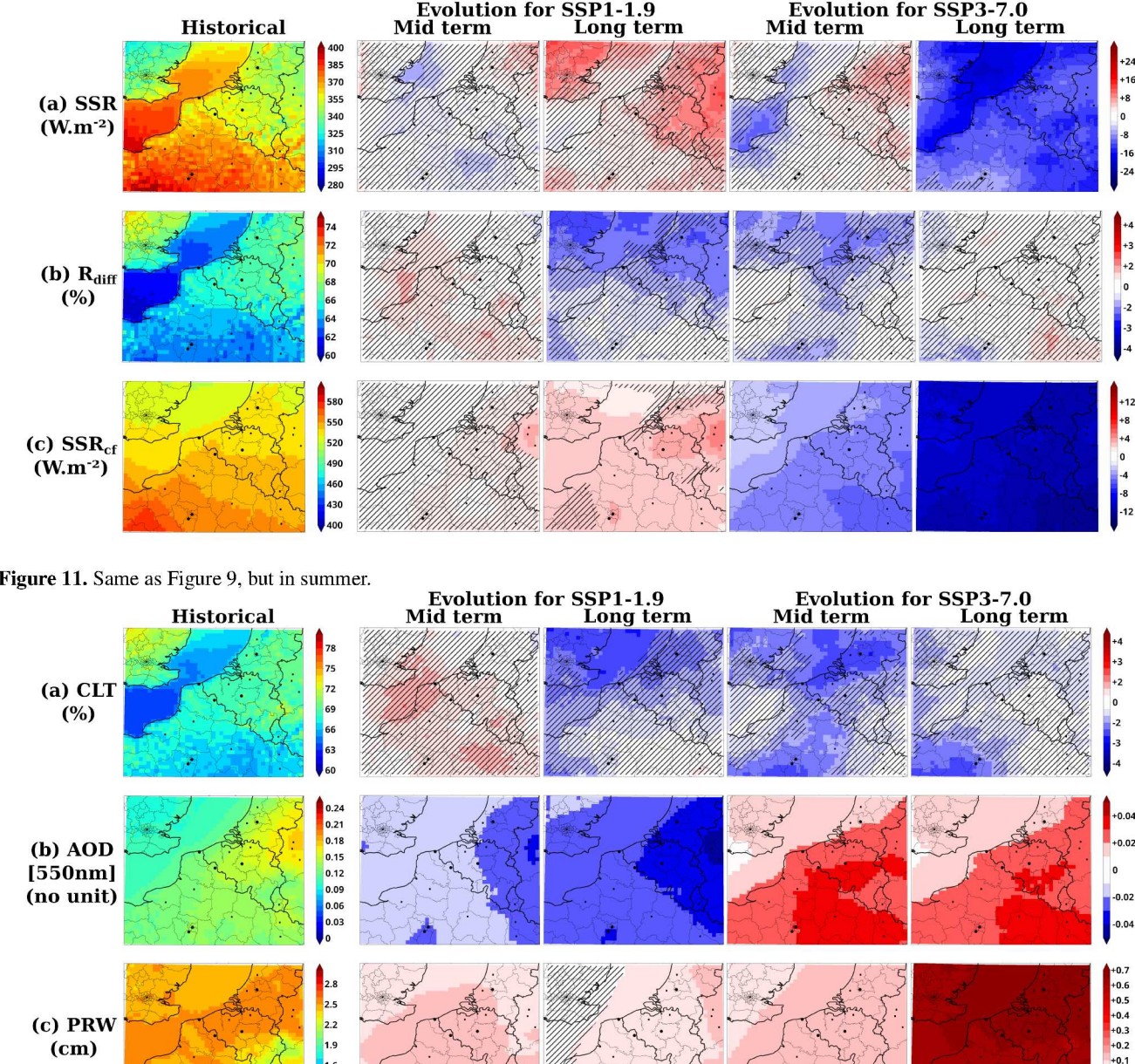

**Figure 11.** Same as Figure 9, but in summer.

**Figure 12.** Same as Figure 10, but in summer.

lower decrease than in spring, especially in 2050. Some areas, like the south-west part of the BNF region in 2100, even show relatively weak or not statistically significant decreases in summertime all-sky SSR (Fig. 11a), despite overall pronounced and expanded decreases in cloud-free SSR (Fig. 11c), much stronger than those predicted in spring for the same SSP3-7.0 scenario (Fig. 9c).

Overall, ALADIN simulations for SSP1-1.9 highlight spatially limited and moderate increases of the all-sky SSR for both spring and summer seasons, with significant changes only over the south-west of England and some eastern parts of the BNF region in the range +5 to +10 $W.m^{-2}$. These increases appear at the end of the century in link with decreases in AOD, in addition to that of the cloud fraction, which is mainly significant in summer. These summer evolutions, although more moderate and spatially limited, are in agreement with those obtained by Hou et al. (2021) based on GCM simulations over Europe for SSP1-2.6. Especially, these authors evidenced a joint decrease in cloud cover and aerosol loads, which contributes to moderate increases in both clear-sky and all-sky SSR. In contrast, ALADIN simulations for SSP3-7.0 suggest large decreases in all-sky surface solar radiation for both spring and summer seasons with an earlier start (2050) and more pronounced changes in spring than in summer (2100). These decreases are driven by the combined effects of aerosols and water vapor increases for both seasons. They are enhanced in spring by an additional increase of +3 to +4% in cloud fraction over most of the BNF region, which induces maximum SSR decreases reaching -25 $W.m^{-2}$ over Benelux in 2100. In contrast, in summer some north and south parts of the BNF region show cloud fraction decreases in 2050, which could largely counterbalance the effect of increased AOD and PRW. These results oppose with the projections obtained from global climate models over Europe considering scenario SSP5-8.5 (Hou et al., 2021), that suggest a future increase of summer SSR over large parts of Europe by the end of the century.

## 5 Conclusions

Our analysis of current and future possible spatiotemporal evolutions of surface solar radiation together with main associated atmospheric parameters relies on regional climate simulations at high-resolution (1-hour model output at 12.5 x 12.5 km) from ALADIN model. The study is focused on the Benelux/Northern France (BNF) region, a dense urbanized and industrialized part of northwestern Europe. This area is frequently under the influence of cloudy conditions and also impacted by anthropogenic particulate pollution events, with large contributions of nitrate aerosols. We examine the variability of all-sky and clear-sky surface solar radiation (SSR), in link with that of cloudy conditions, aerosols loads, and precipitable water.

A first dataset of hindcast simulations, driven by the ERA-5 reanalysis, allows the evaluation of simulated clear-sky frequency, clear-sky and all-sky SSR, as well as aerosol loads and chemical composition by comparison to coincident ground-based measurements available within the BNF region over the period 2010-2020. This regional evaluation shows reasonable agreement regarding the depiction of intra and inter-annual variability for all variables investigated, despite some moderate overestimates of the aerosol optical depth (AOD) in spring, due to an overestimation of nitrate aerosols, which are predominant within the study area. Especially in spring, their current mean contribution to the AOD is estimated to be around 50%. Conversely, ALADIN tends to underestimate the AOD in summer and winter, most probably due the absence of secondary an-

thropogenic organic aerosols within the TACTIC aerosol scheme. Further analysis of 2010-2020 hindcast simulations in spring and summer at the regional scale also shows a pronounced spatial variability in all-sky SSR over the BNF region. In addition to the expected north-south gradient related to the latitude (i.e., solar geometry), ALADIN simulations display higher SSR values (> 330 $W.m^{-2}$ in summer) over the English Channel and North Sea, together with lower cloud fractions and reduced aerosol loads. In contrast, the northernmost continental parts of the BNF region generally exhibit lower SSR values, influenced by either higher cloud cover as over the south west of England, or elevated aerosol load as over the Benelux region.

In addition, three datasets of regional climate simulations, driven by the global CNRM-ESM2-1 Earth model, allow to assess the future evolution in spring and summer of SSR and associated atmospheric parameters at mid-term (2045-2054) and long-term (2091-2100) horizons, compared to the recent historical period of 2005-2014, for two contrasted CMIP6 scenarios (SSP1-1.9 and SSP3-7.0). ALADIN projections for the optimistic SSP1-1.9 scenario only show some moderate increases of all-sky SSR in 2100 limited to specific areas, despite a generalized decline in European anthropogenic emissions of aerosols and precursors, which induces a decrease of AOD, up to -0.04, over the BNF region. Indeed, although this decrease in AOD leads to an overall significant increase in cloud-free SSR over most of the BNF region in 2100, the increase is somewhat attenuated and spatially limited when considering all-sky SSR. This may be explained in summer by the additional contribution of an increase in precipitable water, that partly counterbalances the AOD decrease. Nonetheless, for the SSP1-1.9 scenario, the summer decrease of the cloud fraction, mostly significant in 2100 over the northwestern parts of the BNF region, probably enhances the effects of aerosols, leading to a moderate increase in all-sky SSR over the south-west of England. For the more pessimistic SSP3-7.0 scenario, ALADIN simulations show large significant decreases of all-sky SSR over most of the BNF region, visible from 2050 in spring and around 2100 in summer. These evolutions are associated to significant increases in both AOD, driven by ammonium and nitrate aerosols, and precipitable water in spring and summer, which contributes to a widespread decrease of cloud-free surface solar radiation over the BNF region. In spring, the effects of aerosols and water vapor increases are reinforced by a large increase in cloud fraction over most of the BNF region, leading to a maximum diminution of the all-sky SSR during this season, with values reaching up to –25 $W.m^{-2}$ in 2100. In contrast, ALADIN predicts a moderate decrease of the cloud fraction during the summer season over the northern and south-western parts of the BNF region in 2050, which tends to mitigate the combined effects of aerosols and water vapor, resulting in overall smaller future changes in all-sky SSR than in spring.

Given the satisfactory performances of ALADIN simulations and the contrasted projected evolutions of the surface solar radiation according to CMIP6 scenarios, the coupling of the model outputs with a solar photovoltaic power production model, as in Jerez et al. (2015) and Gutiérrez et al. (2020), represent a compelling perspective for future works. Such an approach should enable further progress in our understanding of the future potential of the BNF region in terms of solar photovoltaic energy production, and its evolution according to future socio-economic and climatic trajectories. In addition, considering that our results highlight the importance of cloud cover changes in modulating forthcoming aerosol influences on SSR, further investigations of cloud cover evolution and impacts on SSR variability would be highly recommended (Norris and Wild, 2007; Correa et al., 2024).

*Code and data availability.* Model outputs and ground measurements used for the main figures are available at https://doi.org/10.5281/ zenodo.10993193. Python codes to analyze data and generate figures are available from the first author upon request. The AERONET datasets were downloaded from NASA AERONET website https://aeronet.gsfc.nasa.gov/. The irradiance measurements from the ATOLL (Atmospheric Observations in LiLLE, https://www.loa.univ-lille.fr/observations/plateformes.html?p=lille) platform are available from an Easy System Data Repository in Chesnoiu et al. (2024a), along other parameters including notably aerosol and gas column properties from AERONET. Irradiance datasets in Cabauw and Palaiseau were downloaded from the BSRN website https://bsrn.awi.de/. Aethalometer measurements of equivalent black carbon surface concentrations, as well as in-situ data of $PM_1$ mass surface concentrations performed by the ACSM are available from the EBAS website https://ebas-data.nilu.no/Default.aspx.

*Author contributions.* GC: formal analysis and data handling. IC and NF: conceptualization, funding acquisition and supervision. PN and MM: design and run of the simulations. VR: operation and maintenance of in-situ measurements in Lille. All authors provided input on data analysis shown in the paper. GC and IC prepared the manuscript with contributions from all co-authors.

*Competing interests.* The authors declare no competing interests.

*Acknowledgements.* The authors would like to thank all principal investigators of the AERONET and BSRN networks and their staff for establishing and maintaining the different sites used in this study, especially BSRN stations' scientists Wouter Knap (Cabauw, Netherlands) and Jordi Badosa (Palaiseau, France). We also acknowledge Frederique Auriol, Diane Catalfamo and Isabelle Jankowiak (LOA) for their contributions in maintaining surface solar irradiance measurements at ATOLL. We also thank Emmanuel Tison (IMT Nord Europe) for his technical support on aerosol in situ measurements at ATOLL. We also appreciate the comments of the two anonymous reviewers that have helped us to improve the paper.

This study is issued from the work of G. Chesnoiu during his PhD thesis financed by the ADEME and Région Hauts-de-France. Additional funds were granted under the LEFE (Les Enveloppes Fluides et l'Environnement)/IMAGO (Interactions Multiples dans l'Atmosphère, la Glace, et l'Océan) CNRS-INSU program over the period 2021-2023. IMT Nord Europe and LOA acknowledge financial support from the Labex CaPPA project, which is funded by the French National Research Agency (ANR) through the PIA (Programme d'Investissement d'Avenir) under contract ANR-11-LABX-0005-01, and the CLIMIBIO and ECRIN projects, all financed by the Regional Council "Hauts-de-France" and the European Regional Development Fund (ERDF). The ATOLL site is part of the ACTRIS ERIC as one of the French National Facilities for aerosol in situ measurements. It contributes to the CARA program of the LCSQA funded by the French Ministry of Environment.

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
