# Peer review of "Spatial variability and future evolution of surface solar radiation, aerosols and cloud cover over Northern France and Benelux: a regional climate model approach"

_EGUsphere, 2024_

## Author Comment (AC1)

Dear Referee #1,

Thank you for your suggestions and remarks. Consideration of these comments has helped to improve the manuscript. Below you will find the answers to each comment.

**General comments**

*The manuscript "Spatial variability and future evolution of surface solar radiation over Northern France and Benelux: a regional climate model approach" evaluates the performance of the regional climate model CNRM-ALADIN64 comparing the model outputs of different variables with observed values. After the conclusion that the model shows reasonable agreement with observational data, the authors use the output of the model to investigate the spatial variability of surface solar radiation (SSR) in the Benelux region in the recent past, and for two different future scenarios in the mid term and long term. The manuscript is mostly well organized, makes use of valid methods using mostly resonable assumptions and address relevant scientific questions within the scope of the journal. At the present form, I think there are still several points that could be improved or better discussed, but I believe it could be suited for publication after major revisions*

*For the analysis of the future evolution of SSR the authors compare the average values of the historical period to mid term and long term scenarios, rather than showing a time series or depicting the SSR change in terms of "trends per decade", as done by most studies in the topic. However, the approach in the study is also perfectly valid and comprehensible, given that special focus is given to the spatial contrasts. For this analysis, the authors focused only in the spring and summer months. I understand the reasoning on why the authors chose to focus on these months, but I do not think this is the best approach if the intent is to discuss the whole picture of future evolution, as I expected from the title of the manuscript. I also believe some improvement could be done in the evaluation of simulated surface solar radiation (section 3.2). At the present form of the manuscript the evaluation is done in terms of the monthly mean absolute SSR values only, and that is dominated by the seasonality. Therefore, this does not provide a strong tool for evaluation of other components of the time series, such as trend and irregular variability (anomalies). In the specific comments I comment in more details these and other minor concerns.*

**Specific comments**

*Lines 22-23: here you basically mention the three major aspects that affect long term SSR variability. I don't know if there is space in the abstract, but I*

*would include a few words to just highlight if each of them is more relevant than the other.*

The end of the abstract has been modified to take into account this comment.

The new sentences describe more accurately our results (see lines 20-25 of the revised manuscript):

"The reductions are maximum in spring due to combined effects of higher cloud cover and nitrate aerosol increases over the Benelux starting in 2050. In summer, cloud cover upcoming decreases largely attenuate the reduction of SSR due to aerosols in 2050, while in 2100 this attenuation is offset by strong water vapor increases. Thus, this regional climate model approach highlights seasonally and spatially variable impacts of future anthropogenic aerosol emissions on SSR evolution over the 21st century. Indeed, over this part of western Europe, cloud cover and water vapor modifications will likely largely contribute to modulate forthcoming aerosol influences."

*Lines 34-45: My understanding of the structure of the introduction was: first paragraph - SSR is important and presents variability; second paragraph - previous literature on SSR variability; third paragraph - puts the manuscript and the region of study in the context; and fourth paragraph - structure of the manuscript. If this is more or less your intention, I think the second paragraph (lines 34-45) could present more literature that discusses SSR changes and their causes in Europe or optimally in or around the region of study. Maybe the review paper by Wild (2009) [https://doi.org/10.1029/2008JD011470] is a good start, but newer papers should be available too. At the moment I have the feeling that this paragraph discussed only projecting SSR, but before talking about the ability to make accurate projects of SSR, it is important to discuss the physical processes too.*

Following the referee's advice, the second paragraph of the introduction has been modified to include more literature about past SSR changes and their causes, especially in Europe (see lines 38-49 of the revised manuscript).

The added paragraph is included below:

"Numerous studies conducted in recent decades, utilizing both observations and modeling, indicate that surface solar radiation has been subject to significant decadal variations in the past, with a worldwide decreasing (dimming) trend until the 1990s, and conversely an increasing (brightening) trend from then onwards (Wild et al., 2005; Wild, 2009; Liepert, 2002; Norris and Wild, 2007). Determining the causes of these trends has been challenging due to the complex interplay of various forcing agents, which directly affect SSR variability through scattering and absorption, and also alter atmospheric dynamics and cloud formation. In Europe, an increasing

number of studies suggest that the rise in all-sky radiation since around 1985 is attributable to changes in cloud cover and anthropogenic aerosol emissions (Schwarz et al., 2020; Boers et al., 2017; Dong et al., 2022; Wild et al., 2021). However, it should be noted that due to the high spatial and temporal variability of clouds and aerosols, their influence and the resulting trends in SSR, exhibit strong spatiotemporal variations that require further investigation. Pfeifroth et al. (2018) notably show that, over 1992-2015, maximum positive trends occur in spring and autumn across western, central and eastern Europe, whereas the winter season and southern Europe exhibit weaker increases or even negative trends in SSR."

*Figure 3: Would be interesting to also include a third line (or a bar) in each plot showing the difference between the two different estimates.*

A bar plot has been added to each panel of Figure 3 to represent the monthly difference between the two estimates (see Figure 3 of the revised manuscript). Additionally, complementary comments regarding the absolute differences between these estimates have been added to the text (see lines 320-323 of Section 3.1.2).

Similar changes have also been applied to Figures S1, S2 and S3 (now Figures S2, S3 and S4) as they are quite similar to Figure 3.

*Section 3.2: The evaluation of the simulated SSR is made by comparing the monthly mean absolute values of direct, diffuse and global components of SSR. However, I do not think this is a complete way of evaluating the SSR representation by the model. For me, the actual evaluation shows that the model can well capture the seasonality. Which is good. But if the model will be used to investigate the future evolution of SSR, we have to also be able to evaluate the model's ability to represent trends and anomalies. For example, in line 332 the authors mention that ALADIN underestimates the direct component in spring by around -20 to -40 W/m2. Such an underestimation represents less than 10% for the absolute SSR of those months, but such a difference could have a much bigger relevance if we would be evaluating in terms of anomalies. Therefore, I think it is important to somehow evaluate the time series with its different components, not only the seasonality.*

In response to the referee's comment a new section (Section 3.4), titled "Monthly time series over 2010-2020", has been added to the revised manuscript (see pages 18 to 20).

This section describes comparisons of the monthly time series of daytime (a) clear-sky frequency, (b) all-sky surface solar radiation and (c) aerosol optical depth at 550 nm in Lille over the period 2010-2020. As the period 2010-2020 is too brief to precisely assess ALADIN's ability to represent observed trends, this section focuses on the analysis of monthly and seasonal anomalies recorded during 2010-2020.

Comparisons of aerosol surface concentrations are also excluded as $PM_1$ ground measurements are only available in Lille from 2016 to 2020.

For brevity, only comparisons in Lille are presented in this section (Figures 7a-c, available below), however similar analyses and figures for Cabauw and Palaiseau are available in the supplements (Figs.S4 and S5, respectively, also shown below).

Overall, the results presented in this section highlight that despite specific seasonal biases discussed in Sections 3.1, 3.2 and 3.3, ALADIN simulations correctly transcribe clear-sky frequency, all-sky SSR and AOD inter-annual variabilities over the period 2010-2020.

Our analysis also suggests ALADIN overestimations of the AOD in spring appear to be more frequent over recent years, especially in Lille (Fig. 7c) and Palaiseau (Fig. S6c). This behavior could be related to the general decreasing trend in AOD observed for several European AERONET  stations since 1995 (Ningombam et al., 2019), which the model may not capture accurately. Further investigations over longer periods, which are out of the scope of this article, are required to provide more robust conclusions on this topic.

Below are the time series added to the revised version of the manuscript (Figures 7, S4 and S5, respectively).

In addition, a few words have been included in the conclusions (see line 631 of the revised manuscript).

**Reference :** Ningombam, S. S., Larson, E., Dumka, U., Estellés, V., Campanelli, M., and Steve, C.: Long-term (1995–2018) aerosol optical depth derived using ground based AERONET and SKYNET measurements from aerosol aged-background sites, Atmospheric Pollution Research, 10, 608–620, https://doi.org/10.1016/j.apr.2018.10.008, 2019.

[Figure]

[Figure]

*Line 401-405: Here the authors justify the use of only spring and summer months in the analysis. First the framing of the argument sounds odd: "our analysis of ground measurements and ALLADIN simulations reveals that spring and summer are both characterized by relatively high SSR values". That is absolutely correct, but it is framed as if this is a new finding from the analysis, while in fact, relatively high SSR values in summer months are simply a fact for locations at such latitude. Furthermore, omitting half of the year when analyzing future evolution of SSR might lead to the false assumption that the this half of the year is not relevant to the long term SSR variability. In studies like Stern et al. (2009 [https://doi.org/10.1002/joc.1735]) and Schilliger et al. (2024 [https://doi.org/10.22541/essoar.171136948.88001430/v1]) the authors show contributions from different months for the SSR trends, and in some cases it is possible to identify significant contributions to the long term SSR trends from processes occurring in the winter months. However, I think there are a few alternatives to address this issue. The most obvious, but probably requiring more work, would be to perform the analysis for all four seasons. But an alternative could also to make it more explicit from the beginning (maybe even in the title) that the study analyses only spring and summer months, because the title and the manuscript until this points led me to expect something like an analysis of the entire year. Or even, another alternative could be to include an analysis of annual mean conditions. Maybe the annual conditions follow*

*very closely the spring and summer conditions for this regions, and this could make it clear for the reader. In any case, if not all months are included in the analysis, it would be important to discuss that long term SSR changes are not the result of only spring and summer months.*

Indeed, the phrasing of lines 401-405 of the initial manuscript was a bit odd and not particularly well positioned. These lines have been removed. Instead, following the referee's advice, the justifications for our focus on spring and summer seasons are now discussed directly at the end of the introduction (see lines 87-91 of the revised text), as follows:

"In order to allow the evaluation of the influence of aerosol anthropogenic emissions future trajectories on SSR evolution, we focus on spring (i.e. March-April-May) and summer (i.e. June-July-August) seasons. Over the Benelux/Northern France region, both aerosol loads and SSR are maximum at this period of the year, coincidently with more frequent clear-sun conditions, as shown by ground-based measurements analysis (Chesnoiu et al., 2024b)."

The aim of this paragraph is to make clear and justify from the beginning that our analysis of the present and future spatiotemporal variability of the SSR and associated parameters is limited to spring and summer seasons due to their particular irradiation conditions, characterized notably by higher SSR levels, lower cloud influence, and higher aerosol loads, which facilitate the evaluation of the influence of aerosol anthropogenic emissions on SSR variability.

Note that in response to a comment from the second referee, a few sentences have been included in the main text of Section 4.1 (see lines 483-489) to highlight similarities and discrepancies between the spatial variability of the SSR and associated parameters in spring/summer seasons (previously Fig. 7, now Fig. 8) and winter/autumn (Fig. S7). Overall, comparisons between Figures 8 and S7 show similar spatial patterns albeit generally lower AOD and SSR values, and higher cloud fractions. This further asserts the focus of our study on spring and summer seasons due to their particular atmospheric conditions.

*Line 455: Is it possible to estimate the effect of this 2-4% cloud fraction change on SSR values?*

There are two possible approaches that could be employed to perform such an analysis.

Similarly to our approach for AOD and PRW (see lines 517-528 of the revised manuscript), SSR sensitivity to changes in cloud fraction could be used to estimate the effect of the cloud fraction changes on SSR values. However, to the best of our knowledge, no studies from the literature have yet investigated the specific sensitivity of SSR to changes in cloud fraction, making it difficult to apply this approach within the scope of our current work.

Another approach would be to use the model's simulations of SSR and SSRcf to derive the overall influence of clouds. This, however, encompasses the influence of not only the cloud fraction, but also the cloud optical depth, and other cloud optical properties.Distinguishing the influence of cloud fraction from other cloud-related parameters, such as cloud optical depth, would require additional data on cloud properties and SSR sensitivity to each parameter—data that was not available at the time of our study.

Furthermore, as now clearly stated in the introduction (lines 87-91 of the revised text), our study focuses on spring and summer seasons due to their particular irradiation conditions, characterized by higher SSR levels, lower cloud influence, and higher aerosol loads, that facilitate the evaluation of the influence of aerosol anthropogenic emissions on SSR variability. Therefore, while the quantification of the cloud fraction's impact on SSR is an interesting prospect for future works, it was deemed beyond the scope of our study, which specifically aims to characterize the influence of aerosols on SSR variability during seasons with high aerosol loads and lower cloud influence.

***Figures 8 and 9: Would be nice to include (maybe even only in the legend of the figures) to what period (years) the mid term and long term refer to.***

The following sentence has been added to both figures 8 and 9 (see Figures 9 and 10 of the revised manuscript):

"The designation "Mid term" represents future changes projected under each scenario for the period 2045-2054, while "Long term" refers to the period 2091-2100."

***Table 3 a): The sum of each component of the SSP1-1.9 scenario does not correspond to the total value displayed in the table. I think there is one zero missing in the value for sulfate.***

Indeed, zeros were missing for the mean AOD at 550 nm reported for the sulfate aerosols under SS1-1.9 in spring. Thank you for noticing. The manuscript has been changed accordingly (see Table 3a of the revised manuscript, page 23).

***Lines 448-482: This paragraph was a little bit hard to read, especially the second half of it. It is also very long. Maybe would be good to reorganize in shorter paragraphs trying to be clearer.***

Lines 448-482 have been modified for clarity (see lines 499-532 of the revised manuscript).

The paragraph has been reorganized in shorter paragraphs.

Contextual formulations (e.g. springtime, summertime, for SSPX-Y scenario, etc.) have been added to facilitate comprehension of the paragraph.

Also, this paragraph has been clarified following the next comment.

The modified paragraph is transcribed below (changes are in bold text):

[revised manuscript text omitted]

*Lines 469-470: The authors refer to the change in SSR that would be induced by some changes in AOD. Is this the change that was observed in the scenarios simulated? This was not too clear.*

The changes in SSR induced by changes in AOD discussed lines 469-470 are not directly simulated by the model. Instead, these changes are derived by multiplying future changes in AOD projected by ALADIN and mean sensitivities of cloud-free SSR to various parameters (e.g. AOD, PRW, SSA, etc.) from Chesnoiu et al. 2024b. The benefit of this approach is that it can be applied to future changes in AOD (lines 469-470) as well as changes in PRW (lines 474-475).

Overall, as mentioned lines 475-477 of the initial manuscript: "*These estimates*" (i.e. changes in SSR$_{cf}$ estimated using the method described above) "*are consistent with ALADIN simulations of SSR$_{cf}$ (Figures 8c) under SSP3-7.0, with an overall decrease of about -3 to -7 W/m² in 2050 and -8 to -12 W/m² in 2100, reflecting the combined effects of aerosols and water vapor.*" This approach has also been used to quantify the contribution of changes in AOD and PRW to the future evolution of SSR$_{cf}$ for SSP1-1.9 (lines 504-506 of the initial manuscript), and gave satisfactory results.

For clarity, lines 468-477 have been changed (see lines 517-528 of the revised manuscript) as shown in the previous response.

*Conclusions: If any of the major concerns raised here are somehow addressed, it would be important to discuss them in this section too.*

It is true that developments related to the comments of both referees should be addressed in the conclusions.

As the focus on the spring and summer seasons is now properly defined in the introduction, no particular comments have been added in the conclusions regarding the analysis of winter and autumn seasons.

In summary, only a few words have been included relative to the analysis of monthly anomalies over the period 2010-2020 (Section 3.4) added in the revised version of Section 3 (see line 631 of the revised manuscript).

*Technical comments*

*Line 27: This first sentence sounds odd. I would change "should not be considered stable over past and upcoming decades" to something like "has not been stable over the past decades and should not be expected to be in the upcoming decades".*

The sentence has been corrected as advised (see lines 28-29 of the revised manuscript).

*Line 34: This first sentence (before comma) sounds confusing, I had to reread to understand. Maybe replace "energy transition that requires increases of pv technology deployment" with something shorter or more objective as simply "PV energy production" or "PV energy production (relevant for energy transition)".*

Following the referee's advice, the sentence has been changed as follows (see lines 36-37 of the revised manuscript, changes are in bold text):

"In the context of climate change that requires **an** increase of photovoltaic **energy production (relevant for  the energy transition)**,[...]"

*Line 53: By "cloudy sky conditions" you meant "cloud cover"?*

Yes. The term "cloudy sky conditions" has been changed to "cloud cover" for clarity (see line 66 of the revised manuscript).

*Line 64: "Cloudy conditions" or "cloud cover"? Maybe you want to use a synonym of cloud cover, such as "cloudiness conditions".?*

We meant "cloud cover". The term "cloudy conditions" has been changed to "cloud cover" for clarity (see lines 77 of the revised manuscript).

***Line 170: "Scenarios is" - either "scenario is" or "scenarios are". Maybe also double check this throughout the manuscript.***

There was an extra "s". The appropriate spelling should read "the SSP1-1.9 scenario is [...]". The text has been modified accordingly (see line 187 of the revised text) and the rest of the manuscript has been double checked.

***Line 227: Maybe also good to include the relative (%) values for the AOD uncertainty.***

Careful review of reference AERONET publications suggests that there is no average relative uncertainty value for AERONET AOD measurements.

From the most recent publication of Giles et al. (2019) regarding version 3 of the AERONET database:

"***Bias and uncertainty estimates for near-real-time AOD are computed by using the difference of the pre-field calibration AOD minus the interpolated calibration AOD. The near-real-time AERONET data have an estimated bias of up to +0.02 and 1σ uncertainty of up to 0.02; these values have slightly higher uncertainty for shorter wavelengths and slightly lower uncertainty for longer wavelengths.***"

This is in accordance with the uncertainty given line 227 of the initial manuscript.

**Reference:** Giles, D. M., Sinyuk, A., Sorokin, M. G., Schafer, J. S., Smirnov, A., Slutsker, I., Eck, T. F., Holben, B. N., Lewis, J. R., Campbell, J. R., Welton, E. J., Korkin, S. V., and Lyapustin, A. I.: Advancements in the Aerosol Robotic Network (AERONET) Version 3 database – automated near-real-time quality control algorithm with improved cloud screening for Sun photometer aerosol optical depth (AOD) measurements, Atmospheric Measurement Techniques, 12, 169–209, https://doi.org/10.5194/amt-12-169-2019, 2019.

***Line 485: Same as line 170.***

The extra "s" has been removed. The sentence now reads "significant for this scenario due to [...]" (see line 536 of the revised manuscript).

---

## Author Comment (AC2)

*General comments:*

*Review of "Spatial variability and future evolution of surface solar radiation over Northern France and Benelux: a regional climate model approach"*

*by Gabriel Chesnoiu, Isabelle Chiapello, Nicolas Ferlay, Pierre Nabat, Marc Mallet, and Véronique Riffault*

Dear Referee #2,

Thank you for your suggestions and remarks. Consideration of these comments has helped to improve the manuscript. Below you will find the answers to each comment.

*The authors present results evaluating the performance of the regional climate model CNRM-ALADIN64 comparing the simulations to observations of surface radiative fluxes and aerosols properties focusing on the west-European area over Benelux and Northern France (BNF) and provide historical period, mid-term and long-term future scenarios. The analysis evaluates the simulations from ALADIN hindcast from 2010 to 2020 allowing the comparison of observations at several sites within the BNF region. Their methodology starts with the assessment of clear-sky frequency based on the methodology of Long and Ackerman (2000) method considering the annual cycle variation at three different sites. Then they focus on the analysis of surface solar radiation by later delving into the annual variation of aerosol properties. Then the paper puts the spatial variability into context for the period 2010-2020. Finally, the manuscripts report the future evolution of two scenarios by presenting mean statistics and illustrating the spatial differences.*

*The manuscript has a comprehensible structure, makes use of valid methods and provides mostly well-documented and clear explanations of their assumptions and limitations. I consider the manuscript to be suited for publication after the following revisions are addressed.*

*Major comments*

*Identification of clear-sky situations*

*The overall analysis is clearly described, but not so much is discussed on other implemented methods that could have been used at the mentioned sites. It is not expected to reformulate the methodology but to better justify the selection of the methodology of Long and Ackerman (2020). More discussion*

*is needed. The authors can refer to the following references for example, or any other the authors might see suitable:*

*M.J. Reno, C.W. Hansen, Identification of periods of clear sky irradiance in time series of GHI measurements, Renew. Energy 90 (2016) 520–531, https://doi.org/10.1016/j.renene.2015.12.031.*

*Elias, T., Ferlay, N., Chesnoiu, G., Chiapello, I., and Moulana, M.: Regional validation of the solar irradiance tool SolaRes in clear-sky conditions, with a focus on the aerosol module, Atmos. Meas. Tech., 17, 4041–4063, https://doi.org/10.5194/amt-17-4041-2024, 2024.*

*Al Asmar, L.; Musson Genon, L.; Eric, D.; Dupont, J.C.; Sartelet, K. Improvement of solar irradiance modelling during cloudy sky days using measurements. Sol. Energy 2021, 230, 1175–1188.*

The referee is right, the selection of the Long and Ackerman method needs more discussion since numerous methods exist in the literature for the identification of clear-sky conditions.

Our choice is based on the method's limited number of input parameters (solar zenith angle, global and diffuse SSR) and high adaptability, as it automatically adjusts to the specific conditions of any observational station. This means its application could easily be extended to additional stations where only global and diffuse SSR measurements are available.

In addition, our choice is based on the results of the comparative study of Gueymard et al. (2019), which showed its high precision for the identification of clear-sky conditions. The method notably achieved the second lowest "false positive" score (i.e. percentage of cloudy situations identified as clear-sky) of 7.25%, despite not depending on collocated photometric measurements or clear-sky simulations.

The introductory paragraph of Section 3.1.1 (see lines 279-287 of the revised manuscript) has been modified as follows (changes are in bold text):

"**Although numerous methods have been described in the literature (Reno and Hansen, 2016; Gueymard et al., 2019; Al Asmar et al., 2021), our study relies on** the well-established method of Long and Ackerman (2000)  to distinguish clear and cloudy situations based on high frequency (3 minutes or less) ground measurements of global and diffuse surface solar radiation. **This method, which has been used for numerous studies (e.g. Elias et al. (2024)), was chosen for its limited number of input parameters (solar zenith angle, global and diffuse SSR) and high versatility, as it automatically adapts to the specific conditions of any observational station equipped with measurements of both global and diffuse**

**horizontal irradiances. Our choice is also based on the results of the comparative study of Gueymard et al. (2019), which showed its high precision for the identification of clear-sky conditions. The method notably achieved the second lowest "false positive" score (i.e. percentage of cloudy situations identified as clear-sky) of 7.25%, despite not depending on collocated photometric measurements or clear-sky simulations."**

*Reduction of ammonia*

*In the paper it was mentioned a reduction of 25 % applied to all monthly ammonia emissions. Despite this reduction, nitrate aerosol concentration remain overestimated by the model. Is this due to a parameterization or an assumption within the model?*

As mentioned lines 181-182 of the submitted manuscript, the reduction of 25% of ammonia emissions represents a compromise between reducing the overestimation of the AOD in spring and maintaining realistic nitrate concentrations throughout the rest of the year. It is thus to be expected that the model continues to overestimate the contribution of nitrates in spring.

This overestimation could be linked to the emission inventories used for the simulations, which, despite important efforts from the community, still feature significant uncertainties in ammonia emissions, especially at the local scale (Hoesly et al., 2018).

The overestimation could also be linked to the simplified chemical representation of nitrates within the model described in detail by Drugé et al. (2019). In particular, the variability of nitric acid ($HNO_3$, precursor of nitrate aerosols) defined in ALADIN is based on a fixed monthly climatology taken from CAMS Reanalysis data over 2003-2007, and does not account for the inter-annual variability of the species. Furthermore, due to the low vapour pressure of sulfuric acid (precursor of sulfate aerosols), the formation of ammonium sulfate takes priority over ammonium nitrate formation. Nitrate aerosol concentrations are thus dependent on the variability and uncertainty of sulfuric acid, as the nitric acid can only interact with the ammonia that remains after formation of sulfate aerosols.

*References:*

Hoesly, R. M., Smith, S. J., Feng, L., Klimont, Z., Janssens-Maenhout, G., Pitkanen, T., Seibert, J. J., Vu, L., Andres, R. J., Bolt, R. M., Bond, T. C., Dawidowski, L., Kholod, N., Kurokawa, J.-I., Li, M., Liu, L., Lu, Z., Moura, M. C. P., O'Rourke, P. R., and Zhang, Q.: Historical (1750–2014) anthropogenic emissions of reactive gases and aerosols from the Community Emissions Data System (CEDS), Geoscientific Model Development, 11, 369–408, https://doi.org/10.5194/gmd-11-369-2018, 2018.

Drugé, T., Nabat, P., Mallet, M., and Somot, S.: Model simulation of ammonium and nitrate aerosols distribution in the Euro-Mediterranean region and their radiative and climatic effects over 1979–2016, Atmospheric Chemistry and Physics, 19, 3707–3731,https://doi.org/10.5194/acp-19-3707-2019, 2019.

***Can the authors comment how future work should consider similar/higher reduction?***

The chosen reduction factor is strongly influenced by the selected emission inventories, the specific parametrization defined in the model for the formation of nitrate aerosols, and the specific location of the simulations. The Benelux/North of France region is largely impacted by agricultural activities and corresponding ammonia emissions, and the influence of these parameters can significantly fluctuate between regions of the world. Hence, we cannot certify that the reduction used in the present study can be generalized.

We recommend that future studies involving other models, regions or study areas carefully assess nitrate aerosol simulations, which can have a decisive impact on the accuracy of other simulated parameters such as the AOD and SSR.

***Was a sensitivity analysis made varying the concentration of ammonia? Or should this be recommended?***

In our study, the reduction factor has been the subject of an extensive sensitivity analysis in which different reduction factors as well as several emission inventories have been tested.

This sensitivity analysis has shown that overall a reduction factor of 50% enables a better description of the mean AOD in spring. However, it also highlighted that such a reduction factor leads to an important underestimation of the AOD in summer. In this context, we chose a more reasonable reduction factor of 25%, which gives the best results over the year,  as it represents a compromise between reducing the overestimation of the AOD in spring and maintaining realistic nitrate concentrations throughout the rest of the year.

Note that while a specific 50% reduction factor could potentially be applied only to springtime ammonia emissions for optimal results, this approach is deemed precarious. Therefore, we opted for a consistent correction factor across all months.

Clarifications on the choice of the reduction factor have been added to the text (see lines 197-201 of the revised manuscript). The changes are summarized below (in bold text):

**"The choice of the reduction factor has been the subject of an extensive sensitivity analysis. The retained** adjustment factor **of 25\%, specific to our study,** represents a compromise between reducing the overestimation in spring and maintaining realistic nitrate concentrations throughout the rest of the year. It can be emphasized that such corrections are consistent with current uncertainties in ammonia emissions, which remain significant, especially at the local scale (Hoesly et al., 2018)."

*Spatial variability*

*It is understandable the need to deepen the analysis for Spring and Summer seasons for the spatial variability analysis. However, including the analysis illustrated in Figure 7 for Autumn and Winter will enrich the overall analysis and further interpretation with Figure 3 and Figure 4a.*

In response to the referee's comment the following figure has been added to the supplements (Fig. S7 of the revised manuscript).

In addition, a few comments have been included in the main text of Section 4.1 (see lines 483-489) to highlight similarities and discrepancies between spring/summer seasons (previously Fig. 7, now Fig. 8) and winter/autumn (Fig. S7).

The added comments are transcribed below:

"Corresponding simulations of the spatial variability of SSR and associated atmospheric parameters for winter (i.e. December-January-February) and autumn (i.e. September-October-November) seasons are reported in the supplements (Fig. S7). For these two seasons, the spatial patterns are similar to those observed in spring and summer (Fig. 8). However, as expected, AOD ranges are significantly reduced over most of the BNF region, together with increased simulated CLT levels, from 72% to more than 80%. Simulated SSR are largely reduced, below 150 $W.m^{-2}$ in winter, and 220 $W.m^{-2}$ in autumn. Thus, in order to evaluate the impacts of contrasted anthropogenic aerosol future emissions on the high-end range of SSR, we will focus on spring and summer seasons."

[Figure]

*Technical/Minor comments:*

*Follow ACP guidelines to refer figures in the text. For instance, change (Figure 1) to (Fig. 1).*

All references of figures inside parentheses in the text have been changed according to the ACP guidelines.

*Correct units. They should be written exponentially in the text, tables and figures.*

Units are now written exponentially in the text, tables and figures.

*Homogenize how to address chemical species. For example, nitrate (NO3) is defined more than one time.*

Some chemical species were indeed defined several times.

Definitions of all acronyms, including for chemical species, have been checked and redundancies have been addressed.

*|| = line*

*|| 1 … change spatio-temporal to spatiotemporal as it was done later in the text*

Done.

*|| 34 ... increases in photovoltaic*

Following a comment from the other referee, the sentence has been changed as follows (see line 36 of the revised manuscript, changes are in bold text):

"In the context of climate change that requires **an** increase of photovoltaic **energy production (relevant for  the energy transition)**,[...]"

*|| 40 correct sentence… assessment of aerosols' future evolutions in time or assessment of the future evolution of aerosols in time*

The sentence has been changed as follows (see line 53 of the revised manuscript, changes are in bold text with a green background):

"[...] assessment of aerosols' future evolutions in time,[...]"

*|| 77 ... climate model and the two kinds of*

Following the referee's comment, the sentence line 70 has been changed from "[...] climate model and of the two kinds of [...]" to "[...] climate model and the two kinds of [...]" (see line 83 of the revised manuscript).

*|| 88 … Close parenthesis SURFEX (SURFace EXternalisée, Masson et al. (2013))*

Done.

*|| 148 … A first dataset → The first dataset*

Done.

*|| 168 remove double parenthesis after van Marle et al. (2017)*

Done.

*|| 170 … scenarios are*

There was an extra "s". The appropriate spelling should read "the SSP1-1.9 scenario is [...]". The text has been modified accordingly (see line 187 of the revised text) and the rest of the manuscript has been double checked following comments from both referees.

*|| 171 … greenhouse gases emissions → greenhouse gas emissions*

Done.

*|| 187 BNF region cf Figure 1 → BNF region Fig. 1*

Done.

*|| 210 Could you add a reference in line 210? or specify that to the best of your knowledge you decide to go for those uncertainties.*

The cited uncertainties refer to the results of Vuilleumier et al. (2014) mentioned line 208. However, as the referee pointed out, it wasn't clear.

For clarity, the reference has been added at the end of line 227 of the revised manuscript.

*|| 224 include space ... 550nm → 550 nm*

Done.

*|| 269 CLT already defined in line 247*

Following a previous comment of the referee, definitions of all acronyms have been homogenized. The iteration of "CLT" line 269 has been removed (see line 297 of the revised manuscript).

*|| 388 The comparisons shown in Figure 6b*

The preposition "in" has been added as advised by the referee (see line 416 of the revised manuscript).

*|| 388:390 Improve clarity. The sentence is too long. 'The comparisons shown in Figure 6b highlight that the underestimation of organic and black carbon aerosols is partially offset in spring and summer by a coincident overestimation of nitrate aerosol concentrations, especially in March and April (around +2 µg/m3), despite the application of a 25% correction factor on ammonia emissions, the main precursor of nitrate aerosols.'*

The sentence was indeed too long. It has been changed as follows (see lines 416-419 of the revised manuscript, changes are highlighted in bold text):

"The comparisons shown **in** Figure 6b highlight that the underestimation of organic and black carbon aerosols is partially offset in spring and summer by a coincident overestimation of nitrate aerosol concentrations, **despite the application of a 25% correction factor on ammonia emissions, the main precursor of nitrate aerosols. This offset is especially significant in March and April with differences in total concentrations of around +2 µg/m3 between the model and the measurements.**"

*|| 410:411 Correct description. Panel (d) is AOD and panel (c) is CLT*

Indeed, the description of the panels was not correct. It has been corrected (see line 452 of the revised manuscript).

*|| 467 Is it BC or equivalent BC?. Keep it consistent along the entire manuscript.*

The species mentioned line 467 is BC. The term "equivalent BC" is generally used in the literature to refer to the concentrations derived from the aethalometer, such as the one used for the evaluation of ALADIN HINDCAST simulations in Section 3.3.

This is why the term "equivalent BC, or eBC", has been initially defined in section 2.2.2 (line 236 of the submitted manuscript). However, although it was not properly stated, the term "eBC" was replaced with more simply "BC" for consistency with ALADIN simulations.

To avoid confusions, this change in terminology is now clearly stated in the revised manuscript (see lines 255-257, Section 2.2.2) as follows:

"For consistency with the terminology of ALADIN simulations, equivalent concentrations of black carbon (i.e. eBC) derived by the aethalometer are hereafter referred to simply as BC."

**|| *534 change spatio-temporal to spatiotemporal***

Done.

**|| *545 AOD already defined in line 219***

To make the article easier to read, we find it preferable to redefine all acronyms within the conclusions, even though they were defined earlier in the text.

***Although it might be obvious, clarify which months the authors consider for their Spring and Summer comparison.***

Spring refers to March-April-May (MAM) and summer to June-July-August (JJA).

Both seasons are now defined in the introduction (line 89 of the revised manuscript).

***Comments on Figures***

***Figure 3, Figure S1, Figure S2 and Figure S3***

***While the lines can be differentiated, the description says green line, but to me it looks blue. Could you change the color of 'Estimate from ALADIN simulations'?***

The color of "Estimate from ALADIN simulations" has been changed to a more noticeable shade of green in Figures 3 (page 12 of the revised manuscript), S1, S2 and S3 (pages 1 and 2 of the revised supplements). In addition, following a comment of the other referee, the monthly mean difference between ALADIN simulations and ground measurements has been added to Figures 3, S1, S2 and S3. For consistency, the color of ALADIN simulations in Figure 6 (page 17) has also been changed.

***Figure 8, 9, 10, 11 Long term evolution for SSP3-7.0 Is it possible to include a separated colorbar for the lowermost right-side panel? In case it messes up the structure of the figure, perhaps include an adequate colorbar for these parameters in the appendix?***

Following the referee's advice, a distinct figure has been added to the revised supplements (see Figure S10, page 7 of the revised supplements). The figure includes a modified version of the lowermost right-side panels of Figures 8-11 (now Figs. 9-12) with a specific colorbar.

---

## Author Response (AR2)

Dear Referee #2, thank you for your feedback on the updated version of the manuscript.

Response to comment #1 :

While we acknowledge that in general performing a direct comparison of the monthly anomalies (10-year monthly means minus monthly values) is an interesting approach, we would like to emphasize that an interannual variability analysis is outside of the main objectives of this paper.
Indeed, our paper focuses on the seasonal and spatial regional variability of surface solar radiation for the current period and for two future climate scenarios at mid- and end-century. All the figures of the paper, except new Figure 7, align with these main topics and for consistency reasons we do not wish, in this paper, to provide a deeper analysis of interannual evolution.

We recall here that in response to your initial comment, we added a fully new section (Section 3.4) and a comprehensive figure (Figure 7) in the revised manuscript. Figure 7 represents all the components of the monthly time series (including seasonality, anomalies, and trends), which allows to provide, in a first approach, the interannual changes while remaining consistent with our study main objectives and scope. Nonetheless, we appreciate the suggestion and will certainly consider this approach as a future reference, as such an analysis would indeed offer additional valuable insights.

Response to comment #2 :

Thank you for the provided references: Norris and Wild (2007) is indeed an interesting approach that deserve to be mentioned. It is included in the references list of the paper.
The very recent article of Correa et al. (2024) is also relevant,with an approach to qualitatively distinguish the effects of changes in cloud cover and cloud optical properties on all-sky SSR, so thanks for the suggestion.

Both references are now mentioned in the conclusions of the manuscript (lines 631-634, page 28).

« In addition, considering that our results highlight the importance of cloud cover changes in modulating forthcoming aerosol influences on SSR, further investigations of cloud cover evolution and impacts on SSR variability would be highly recommended (Norris and Wild, 2007; Correa et al., 2024). »

---

## Author Response (AR3)

Dear Marco Gaetani, thank you for your feedback on the updated version of the manuscript.

In response to your first comment the title has been shortened. However, we believe that the regional climate model approach is an important part of the study, which should be highlighted in the title. Hence we propose the following reformulation of the title (see also the first page of the revised manuscript):

*"Regional modeling of surface solar radiation, aerosols, and cloud cover spatial variability and projections over Northern France and Benelux"*

It comprises 19 words (140 characters) instead of 23 words (161 characters) for the initial title.

In response to your second comment the abstract has also been shortened (now 253 words). The modified text, which can be found on the first page of the revised manuscript, is referenced below :

*"Investigating current and future evolution of surface solar radiation (SSR) is essential in the context of climate change and associated environmental issues. We focus on the influence of atmospheric aerosols, along with cloud cover and water vapor content over northern France and Benelux in spring and summer. Our analysis relies on the CNRM-ALADIN64 regional climate model at 12.5 km resolution, which includes an interactive aerosol scheme. A regional evaluation of 2010-2020 ALADIN hindcast simulations of clear-sky and all-sky SSR, clear-sky frequency and aerosols, by comparison to coincident multi-site ground-based measurements shows reasonable agreement. In addition, these hindcast simulations emphasize how elevated aerosol loads over Benelux and high cloud cover over southwestern England reduce the SSR. Additional ALADIN climate simulations for 2050 and 2100 under CMIP6 SSP1-1.9 predict a significant reduction of aerosol loads compared to 2005-2014, especially over Benelux, associated with future increases in clear-sky SSR but geographically limited all-sky SSR evolution. In contrast, under SSP3-7.0, clear-sky and all-sky SSR is projected to decline significantly over the domain. This decline is maximum in spring over Benelux due to combined increases of cloud cover and nitrate aerosols projected from 2050 onwards. In summer, projected decreases of cloud cover largely attenuate the reduction of SSR due to aerosols in 2050, while by 2100 rising water vapor contents counteract this attenuation. Thus, our results highlight seasonally and spatially variable impacts of future anthropogenic aerosol emissions on SSR evolution, due to cloud cover and water vapor modifications that will likely largely contribute to modulate forthcoming aerosol influences."*